# FIT: A METRIC FOR MODEL SENSITIVITY

**Ben Zandonati**
University of Cambridge
baz23@cam.ac.uk

**Adrian Alan Pol**
Princeton University
ap6964@princeton.edu

**Maurizio Pierini**
CERN
maurizio.pierini@cern.ch

**Olya Sirkin**
CEVA Inc.
sirkinolya@gmail.com

**Tal Kopetz**
CEVA Inc.
tal.kopetz@ceva-dsp.com

## ABSTRACT

Model compression is vital to the deployment of deep learning on edge devices. Low precision representations, achieved via quantization of weights and activations, can reduce inference time and memory requirements. However, quantifying and predicting the response of a model to the changes associated with this procedure remains challenging. This response is non-linear and heterogeneous throughout the network. Understanding which groups of parameters and activations are more sensitive to quantization than others is a critical stage in maximizing efficiency. For this purpose, we propose FIT. Motivated by an information geometric perspective, FIT combines the Fisher information with a model of quantization. We find that FIT can estimate the final performance of a network without retraining. FIT effectively fuses contributions from both parameter and activation quantization into a single metric. Additionally, FIT is fast to compute when compared to existing methods, demonstrating favourable convergence properties. These properties are validated experimentally across hundreds of quantization configurations, with a focus on layer-wise mixed-precision quantization.

## 1 INTRODUCTION

The computational costs and memory footprints associated with deep neural networks (DNN) hamper their deployment to resource-constrained environments like mobile devices (Ignatov et al., 2018), self-driving cars (Liu et al., 2019), or high-energy physics experiments (Coelho et al., 2021). Latency, storage and even environmental limitations directly conflict with the current machine learning regime of performance improvement through scale. For deep learning practitioners, adhering to these strict requirements, whilst implementing state-of-the-art solutions, is a constant challenge. As a result, compression methods, such as quantization (Gray & Neuhoff, 1998) and pruning (Janowsky, 1989), have become essential stages in deployment on the edge.

In this paper, we focus on quantization. Quantization refers to the use of lower-precision representations for values within the network, like weights, activations and even gradients. This could, for example, involve reducing the values stored in 32-bit floating point (FP) single precision, to INT-8/4/2 integer precision (IP). This reduces the memory requirements whilst allowing models to meet strict latency and energy consumption criteria on high-performance hardware such as FPGAs.

Despite these benefits, there is a trade-off associated with quantization. As full precision computation is approximated with less precise representations, the model often incurs a drop in performance. In practice, this trade-off is worthwhile for resource-constrained applications. However, the DNN performance degradation associated with quantization can become unacceptable under aggressive schemes, where post-training quantization (PTQ) to 8 bits and below is applied to the whole network (Jacob et al., 2018). Quantization Aware Training (QAT) (Jacob et al., 2018) is often used to recover lost performance. However, even after QAT, aggressive quantization may still result in a large performance drop. The model performance is limited by sub-optimal quantization schemes.

It is known that different layers, within different architectures, respond differently to quantization (Wu et al., 2018). Akin to how more detailed regions of images are more challenging to compress, as

are certain groups of parameters. As is shown clearly by Wu et al. (2018), uniform bit-width schemes fail to capture this heterogeneity. Mixed-precision quantization (MPQ), where each layer within the network is assigned a different precision, allows us to push the performance-compression trade-off to the limit. However, determining which bit widths to assign to each layer is non-trivial. Furthermore, the search space of quantization configurations is exponential in the number of layers and activations.

Existing methods employ techniques such as neural architecture search and deep reinforcement learning, which are computationally expensive and less general. Methods which aim to explicitly capture the *sensitivity* (or *importance*) of layers within the network, present improved performance and reduced complexity. In particular, previous works employ the Hessian, taking the loss landscape curvature as sensitivity and achieving state-of-the-art compression. Even so, many explicit methods are slow to compute, grounded in intuition, and fail to include activation quantization. Furthermore, previous works determine performance based on only a handful of configurations. Further elaboration is presented in Section 2.

The Fisher Information and the Hessian are closely related. In particular, many previous works in optimisation present the Fisher Information as an alternative to the Hessian. In this paper, we use the **F**isher **I**nformation **T**race as a means of capturing the network dynamics. We obtain our final FIT metric which includes a quantization model, through a general proof in Section 3 grounded within the field of information geometry. The layer-wise form of FIT closely resembles that of Hessian Aware Quantization (HAWQ), presented by Dong et al. (2020). Our contributions in this work are as follows:

1. We introduce the Fisher Information Trace (FIT) metric, to determine the effects of quantization. To the best of our knowledge, this is the first application of the Fisher Information to generate MPQ configurations and predict final model performance. We show that FIT demonstrates improved convergence properties, is faster to compute than alternative metrics, and can be used to predict final model performance after quantization.

2. The sensitivity of parameters and activations to quantization is combined within FIT as a single metric. We show that this consistently improves performance.

3. We introduce a rank correlation evaluation procedure for mixed-precision quantization, which yields more significant results with which to inform practitioners.

## 2 PREVIOUS WORK

In this section, we primarily focus on mixed-precision quantization (MPQ), and also give context to the information geometric perspective and the Hessian.

**Mixed Precision Quantization** As noted in Section 1, the search space of possible quantization configurations, i.e. bit setting for each layer and/or activation, is exponential in the number of layers: $\mathcal{O}(|\mathcal{B}|^{2L})$, where $\mathcal{B}$ is the set of bit precisions and $L$ the layers. Tackling this large search space has proved challenging, however recent works have made headway in improving the state-of-the-art.

CW-HAWQ (Qian et al., 2020), AutoQ (Lou et al., 2019) and HAQ (Wang et al., 2019) deploy Deep Reinforcement Learning (DRL) to automatically determine the required quantization configuration, given a set of constraints (e.g. accuracy, latency or size). AutoQ improves upon HAQ by employing a hierarchical agent with a hardware-aware objective function. CW-HAWQ seeks further improvements by reducing the search space with explicit second-order information, as outlined by Dong et al. (2020). The search space is also often explored using Neural Architecture Search (NAS). For instance, Wu et al. (2018) obtain 10-20× model compression with little to no accuracy degradation. Unfortunately, both the DRL and NAS approaches suffer from large computational resource requirements. As a result, evaluation is only possible on a small number of configurations. These methods explore the search space of possible model configurations, without explicitly capturing the dynamics of the network. Instead, this is learned implicitly, which restricts generalisation.

More recent works have successfully reduced the search space of model configurations through explicit methods, which capture the relative *sensitivity* of layers to quantization. The bit-width assignment is based on this sensitivity. The eigenvalues of the Hessian matrix yield an analogous heuristic to the local curvature. Higher local curvature indicates higher sensitivities to parameter perturbation, as would result from quantization to a lower bit precision. Choi et al. (2016) exploit

this to inform bit-precision configurations. Popularised by Dong et al. (2020), HAWQ presents the following metric, where a trace-based method is combined with a measure of quantization error:

$$\sum_{l=1}^{L} \bar{T}r(H_l) \cdot ||Q(\theta_l) - \theta_l||^2.$$

Here, $\theta_l$ and $Q(\theta_l)$ represent the full precision and model parameters respectively, for each block $l$ of the model. $\bar{T}r(H_l)$ denotes the parameter normalised Hessian trace. HAWQ-V1 (Dong et al., 2019) used the top eigenvalue as an estimate of block sensitivity. However, this proved less effective than using the trace as in HAWQ-V2 Dong et al. (2020). The ordering of quantization depth established over the set of network blocks reduces the search space of possible model configurations. The Pareto front associated with the trade-off between sensitivity and size is used to quickly determine the best MPQ configuration for a given set of constraints. HAWQ-V3 (Yao et al., 2021) involves integer linear programming to determine the quantization configuration. Although Dong et al. (2020) discusses activation quantization, it is considered separately from weight quantization. In addition, trace computation can become very expensive for large networks. This is especially the case for activation traces, which require large amounts of GPU memory. Furthermore, only a handful of configurations are analysed.

Other effective heuristics have been proposed, such as batch normalisation $\gamma$ (Chen et al., 2021) and quantization (?) scaling parameters. For these intuition-grounded heuristics, it is more challenging to assess their generality. More complex methods of obtaining MPQ configurations exist. Kundu et al. (2021) employ straight-through estimators of the gradient (Hubara et al., 2016) with respect to the bit-setting parameter. Adjacent work closes the gap between final accuracy and quantization configuration. In Liu et al. (2021), a classifier after each layer is used to estimate the contribution to accuracy.

Previous works within the field of quantization have used the Fisher Information for adjacent tasks. Kadambi (2020), use it as a regularisation method to reduce quantization error, whilst it is employed by Tu et al. (2016) as a method for computing importance rankings for blocks/parameters. In addition, Li et al. (2021) use the Fisher Information as part of a block/layer-wise reconstruction loss during post-training quantization.

**Connections to the loss landscape perspective**  Gradient preconditioning using the Fisher Information Metric (FIM) - the Natural Gradient - acts to normalise the distortion of the loss landscape. This information is extracted via the eigenvalues which are characterised by the trace. This is closely related to the Hessian matrix (and Netwon's method). The two coincide, provided that the model has converged to the global minimum $\theta^*$, and specific regularity conditions are upheld (Amari, 2016) (see Appendix G.3). The Hessian matrix $H$ is derived via the second-order expansion of the loss function at a minimum. Using it as a method for determining the effects of perturbations is common. The FIM is more general, as it is yielded (infinitesimally) from the invariant f-divergences. As a result, FIT applies to a greater subset of models, even those which have not converged to critical points.

Many previous works have analysed and provided examples for, the relation between second-order information, and the behaviour of the Fisher Information. (Kunstner et al., 2019; Becker & Lecun, 1989; Martens, 2014; Li et al., 2020). This has primarily been with reference to natural gradient descent (Amari, 1998), preconditioning matrices, and Newton's method during optimization. Recent works serve to highlight the success of the layer-wise scaling factors associated with the Adam optimiser (Kingma & Ba, 2014) whilst moving in stochastic gradient descent (SGD) directions (Agarwal et al., 2020; 2022). This is consistent with previous work (Kunstner et al., 2019), which highlights the issues associated with sub-optimal scaling for SGD, and erroneous directions for Empirical Fisher (EF)-based preconditioning. The EF and its properties are also explored by Karakida et al. (2019).

It is important to make the distinction that FIT provides a different use case. Rather than focusing on optimisation, FIT is used to quantify the effects of small parameter movements away from the full precision model, as would arise during quantization. Additionally, the Fisher-Rao metric has been previously suggested as a measure of network capacity (Liang et al., 2019). In this work, we consider taking the expectation over this quantity. From this perspective, FIT denotes expected changes in network capacity as a result of quantization.

## 3 METHOD

First, we outline preliminary notation. We then introduce an information geometric perspective to quantify the effects of model perturbation. FIT is reached via weak assumptions regarding quantization. We then discuss computational details and finally connect FIT to the loss landscape perspective.

### 3.1 PRELIMINARY NOTATION.

Consider training a parameterised model as an estimate of the underlying conditional distribution $p(y|x,\theta)$, where the empirical loss on the training set is given by: $\mathcal{L}(\theta) = \frac{1}{N}\sum_{i=1}^{N} f(x_i, y_i, \theta)$. In this case, $\theta \in \mathbb{R}^w$ are the model parameters, and $f(x_i, y_i, \theta)$ is the loss with respect to a single member $z_i = (x_i, y_i) \in \mathbb{R}^d \times \mathcal{Y}$ of the training dataset $\mathrm{D} = \{(x_i, y_i)\}_{i=1}^{N}$, drawn i.i.d. from the true distribution. Consider, for example, the cross-entropy criterion for training given by: $f(x_i, y_i, \theta) = -\log p(y_i|x_i, \theta)$. Note that in parts we follow signal processing convention and refer to the parameter movement associated with a change in precision as *quantization noise*.

### 3.2 FISHER INFORMATION TRACE (FIT) METRIC

Consider a general perturbation to the model parameters: $p(y|x, \theta + \delta\theta)$. For brevity, we denote $p(y|x, \phi)$ as $p_\phi$. To measure the effect of this perturbation on the model, we use an $f$-divergence (e.g. KL, total variation or $\chi^2$) between them: $D_f(p_\theta||p_{\theta+\delta\theta})$. It is a well-known concept in information geometry (Amari, 2016; Nielsen, 2018) that the FIM arises from such a divergence: $D_{KL}(p_\theta||p_{\theta+\delta\theta}) = \frac{1}{2}\delta\theta^T I(\theta)\delta\theta$. The FIM, $I(\theta)$, takes the following form:

$$I(\theta) = \mathbb{E}_{p_\theta(x,y)}[\nabla_\theta \log p(y|x, \theta)\nabla_\theta \log p(y|x, \theta)^T]$$

In this case, $p_\theta(x,y)$ denotes the fact that expectation is taken over the joint distribution of $x$ and $y$. The Rao distance (Atkinson & Mitchell, 1981) between two distributions then follows.

The exact (per parameter) perturbations $\delta\theta$ associated with quantization are often unknown. As such, we assume they are drawn from an underlying quantization noise distribution, and obtain an expectation of this quadratic differential:

$$\mathbb{E}\left[\delta\theta^T I(\theta)\delta\theta\right] = \mathbb{E}[\delta\theta]^T I(\theta)\mathbb{E}[\delta\theta] + Tr(I(\theta)Cov[\delta\theta])$$

We assume that the random noise associated with quantization is symmetrically distributed around mean of zero: $\mathbb{E}[\delta\theta] = 0$, and uncorrelated: $Cov[\delta\theta] = Diag(\mathbb{E}[\delta\theta^2])$. This yields the FIT heuristic in a general form:

$$\Omega = Tr\left(I(\theta)diag(\mathbb{E}[\delta\theta^2])\right)$$

To develop this further, we assume that parameters within a single layer or block will have the same noise power, as this will be highly dependent on block-based configuration factors. As a concrete example, take the quantization noise power associated with heterogeneous quantization across different layers, which is primarily dependent on layer-wise bit-precision. As such, we can rephrase FIT in a layer-wise form:

$$\sum_{l}^{L} Tr(I(\theta_l)) \cdot \mathbb{E}[\delta\theta^2]_l$$

Where $l$ denotes a single layer/block in a set of $L$ model layers/blocks.

### 3.2.1 EXTENDING THE NEURAL MANIFOLD

The previous analysis applies well to network parameters, however, quantization is also applied to the activations themselves. As such, to determine the effects of the activation noise associated with quantization, we must extend the notion of *neural manifolds* to also include activation statistics, $\hat{a}$, over the dataset as well. The general perturbed model is denoted as follows: $p(y|x, \theta + \delta\theta, \hat{a} + \delta\hat{a})$. The primary practical change required for this extension involves taking derivatives w.r.t. activations rather than parameters - a feature which is well supported in deep learning frameworks (see Appendix

C for activation trace examples). After which, the expectation is taken over the data. Once again, this change is simple, as the empirical form of the Fisher Information requires a similar approximation (Section 3.3, and the two can be computed at the same time. Having considered the quantization of activations and weights within the same space, these can now be meaningfully combined in the FIT heuristic. This is illustrated in Section 4.

### 3.3 COMPUTATIONAL DETAILS

The empirical form of FIT can be obtained via approximations of $\mathbb{E}[\delta\theta^2]_l$ and $Tr(I(\theta_l))$. The former requires either a Monte-Carlo estimate, or an approximate noise model (see Appendix F), whilst the latter requires computing the Empirical Fisher (EF): $\hat{I}(\theta)$. This yields the following form:

$$\sum_l^L \frac{1}{n(l)} \cdot Tr(\hat{I}(\theta_l)) \cdot ||\delta\theta||_l^2 \ .$$

Kunstner et al. (2019) illustrates the limitations, as well as the notational inconsistency, of the EF, highlighting several failure cases when using the EF during optimization. Recall that the FIM involves computing expectation over the joint distribution of $x$ and $y$. We do not have access to this distribution, in fact, we only model the conditional distribution. This leads naturally to the notion of the EF:

$$\hat{I}(\theta) = \frac{1}{N} \sum_{i=1}^N \nabla_\theta f_\theta(z_i) \nabla_\theta f_\theta(z_i)^T \ .$$

The EF trace is far less computationally intensive than the Hessian matrix. Previous methods (Dong et al., 2020; Yao et al., 2021; Yu et al., 2022) suggested the use of the Hutchinson algorithm (Hutchinson, 1989). The trace of the Hessian is extracted in a matrix-free manner via zero-mean random variables with a variance of one: $Tr(H) \approx \frac{1}{m} \sum_{i=1}^m r_i^T H r_i$, where $m$ is the number of estimator iterations. It is common to use Rademacher random variables $r_i \in \{-1, 1\}$. First, the above method requires a second backwards pass through the network for every iteration. This is costly, especially for DNNs with many layers, where the increased memory requirements associated with storing the computation graph become prohibitive. Second, the variance of each estimator can be large. This is given by: $\mathbb{V}[r_i^T H r_i] = 2 \left( ||H||_F^2 - \sum_i H_{ii}^2 \right)$ (see Appendix G.4). Even for the Hessian, which has a high diagonal norm, this variance can still be large. This is validated empirically in Section 4. In contrast, the EF admits a simpler form (see Appendix G.4):

$$Tr[\hat{I}(\theta)] = \frac{1}{N} \sum_{i=1}^N ||\nabla f(z_i, \theta)||^2 \ .$$

The convergence of this trace estimator improves upon that of the Hutchinson estimator in having lower variance. Additionally, the computation is faster and better supported by deep learning frameworks: its computation can be performed with a single network pass, as no second derivative is required. A similar scheme is used in the Adam optimizer (Kingma & Ba, 2014), where an exponential moving average is used. Importantly, the EF trace estimation has a more model-agnostic variance, also validated in Section 4.

## 4 EXPERIMENTS

We perform several experiments to determine the performance of FIT. First, we examine the properties of FIT, and compare it with the commonly used Hessian. We show that FIT preserves the relative block sensitivities of the Hessian, whilst having significantly favourable convergence properties. Second, we illustrate the predictive performance of FIT, in comparison to other sensitivity metrics. We then show the generality of FIT, by analysing the performance on a semantic segmentation task. Finally, we conclude by discussing practical implications.

### 4.1 COMPARISON WITH THE HESSIAN

To evaluate trace performance and convergence in comparison to Hessian-based methods, we consider several computer vision architectures, trained on the ImageNet (Deng et al., 2009) dataset. We first

illustrate the similarity between the EF trace and the Hessian trace and then illustrate the favourable convergence properties of the EF. Further analysis for BERT on SST-2 is given in Appendix A.

**Trace Similarity.**

Figure 1 shows that the EF preserves the relative block sensitivity of the Hessian. Substituting the EF trace for the Hessian will not affect the performance of a heuristic-based search algorithm. Additionally, even for the Inception-V3 trace in Figure 1(d), the scaling discrepancy would present no change in the final generated model configurations because heuristic methods (which search for a minimum) are scale agnostic.

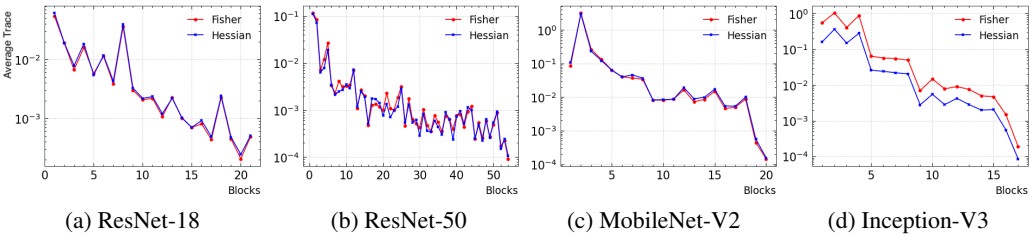

| (a) ResNet-18 | (b) ResNet-50 | (c) MobileNet-V2 | (d) Inception-V3 |

Figure 1: Hessian and EF Parameter traces for four classification models. The Hessian and EF traces for the parameters are very similar. For Inception-V3, this holds up to a constant scaling factor.

**Convergence Rate.**

Across all models, the variance associated with the EF trace is orders of magnitude lower than that of the Hessian trace, as shown in Table 1. This is made clear in Figure 2, where the EF trace stabilises in far fewer iterations than the Hessian. The analysis in Section 3, where we suggest that the EF estimation process has lower variance and converges faster, holds well in practice.

Importantly, whilst the Hessian variance shown in Table 1 is very model dependent, the EF trace estimator variance is more consistent across all models. These results also hold across differing batch sizes (see Appendix D).The results of this favourable convergence are illustrated in Table 1. For fixed tolerances, the model agnostic behaviour, faster computation, and lower variance, all contribute to a large speedup.

| | **Estimator Variance** | | **Iteration Time (ms)** | | **Relative Speedup** |
| | **EF** | Hessian | **EF** | Hessian | |
|---|---|---|---|---|---|
| ResNet-18 | **0.15** $\pm$ 0.03 | 1.09 $\pm$ 0.02 | **47.78** $\pm$ 0.03 | 186.54 $\pm$ 0.56 | **27.67** $\pm$ 5.40 |
| ResNet-50 | **0.31** $\pm$ 0.04 | 6.91 $\pm$ 1.52 | **152.02** $\pm$ 0.38 | 639.13 $\pm$ 1.02 | **94.24** $\pm$ 34.06 |
| MobileNet-V2 | **0.24** $\pm$ 0.01 | 4.81 $\pm$ 0.38 | **58.84** $\pm$ 0.55 | 2573.50 $\pm$ 3.06 | **894.24** $\pm$ 121.25 |
| Inception-V3 | **0.43** $\pm$ 0.03 | 13.62 $\pm$ 0.46 | **235.43** $\pm$ 0.21 | 905.04 $\pm$ 4.69 | **122.06** $\pm$ 14.90 |

Table 1: Representative examples of the typical speedup associated with using the EF over the Hessian. Iteration times and variances are computed as sample statistics over multiple runs of many iterations, with batch size of 32. The resulting speedup is denoted for a fixed tolerance, which can be practically computed via a moving variation of the mean trace. Early stopping is possible when we first reach the desired tolerance. The measurements were performed on an NVidia 2080Ti GPU.

## 4.2 FROM FIT TO ACCURACY

In this section, we use correlation as a novel evaluation criterion for sensitivity metrics, used to inform quantization. Strong correlation implies that the metric is indicative of final performance, whilst low correlation implies that the metric is less informative for MPQ configuration generation. In this MPQ setting, FIT is calculated as follows (Appendix F):

$$\sum_{l}^{L} Tr(\hat{I}(\theta_l)) \cdot \left[ \frac{\theta_{max} - \theta_{min}}{2^{b_l} - 1} \right]^2$$

Each bit configuration - $\{b_l\}_1^L$ - yields a unique FIT value from which we can estimate final performance. The space of possible models is large, as outlined in Section 2, thus we randomly sample

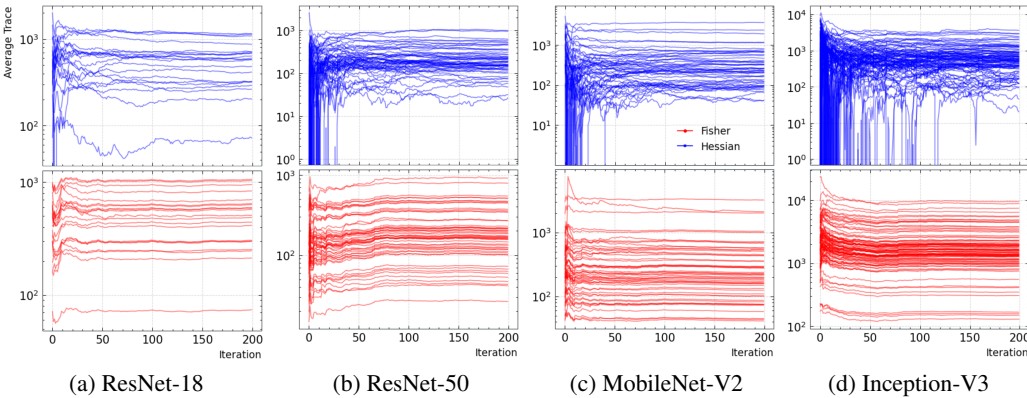

Figure 2: Comparison between Hessian and EF trace convergence for four classification models.

configuration space. 100 CNN models, with and without batch-normalisation, are trained on the Cifar-10 and Mnist datasets, giving a total of 4 studies.

**Comparison Metrics.**

As noted in Section 2 previous works (Kundu et al., 2021; Liu et al., 2021; ?) provide layer-wise heuristics with which to generate quantization configurations. This differs from FIT, which yields a single value. These previous methods are not directly comparable. However, we take inspiration from Chen et al. (2021); ?, using the quantization ranges (QR: $|\theta_{max} - \theta_{min}|$) as well as the batch normalisation scaling parameter (BN: $\gamma$) (Ioffe & Szegedy, 2015), to evaluate the efficacy of FIT in characterising sensitivity.

In addition to these, we also perform ablation studies by removing components of FIT, resulting in $FIT_W$, $FIT_A$, and the isolated quantization noise model, i.e. $\mathbb{E}[\delta\theta^2]$. Note that we do not include HAWQ here as it generates results equivalent to $FIT_W$. Equations for these comparison heuristics are shown in Appendix E. This decomposition helps determine how much the components which comprise FIT contribute to its overall performance.

| Experiment | Dataset | BN | FIT | QR | Noise | $FIT_W$ | $QR_W$ | $FIT_A$ | $QR_A$ | BN |
|---|---|---|---|---|---|---|---|---|---|---|
| A | Cifar-10 | ✓ | **0.89** | 0.76 | 0.85 | 0.87 | 0.86 | 0.38 | 0.36 | 0.33 |
| B | Cifar-10 | | **0.77** | 0.67 | 0.60 | 0.65 | 0.61 | 0.61 | 0.60 | - |
| C | Mnist | ✓ | 0.86 | **0.89** | 0.83 | 0.72 | 0.80 | 0.44 | 0.39 | 0.39 |
| D | Mnist | | **0.90** | 0.58 | 0.70 | 0.72 | 0.72 | 0.55 | 0.44 | - |

Table 2: Rank correlation coefficient for each combination of sensitivity and quantization experiment. W/A subscript indicates using only either weights or activations. BN indicates the presence of batch-normalisation within the architecture.

Figure 3 and Table 2 show the plots and rank-correlation results across all datasets and metrics. From these results, the key benefits of FIT are demonstrated.

**FIT correlates well with final model performance.** From Table 2, we can see that FIT has a consistently high rank correlation coefficient, demonstrating its application in informing final model performance. Whilst other methods vary in correlation, FIT remains consistent across experiments.

**FIT combines contributions from both parameters and activations effectively.** We note from Table 2, that combining $FIT_W$ and $FIT_A$ consistently improves performance. The same is not the case for QR. Concretely, the average increase in correlation with the inclusion of $FIT_A$ is 0.12, whilst for $QR_A$, the correlation decreases on average by 0.02. FIT scales the combination of activation and parameter contributions correctly, leading to a consistent increase in performance. Furthermore, from Table 2 we observe that the inclusion of batch-normalisation alters the relative contribution from parameters and activations. As a result, whilst FIT combines each contribution effectively and remains consistently high, QR does not.

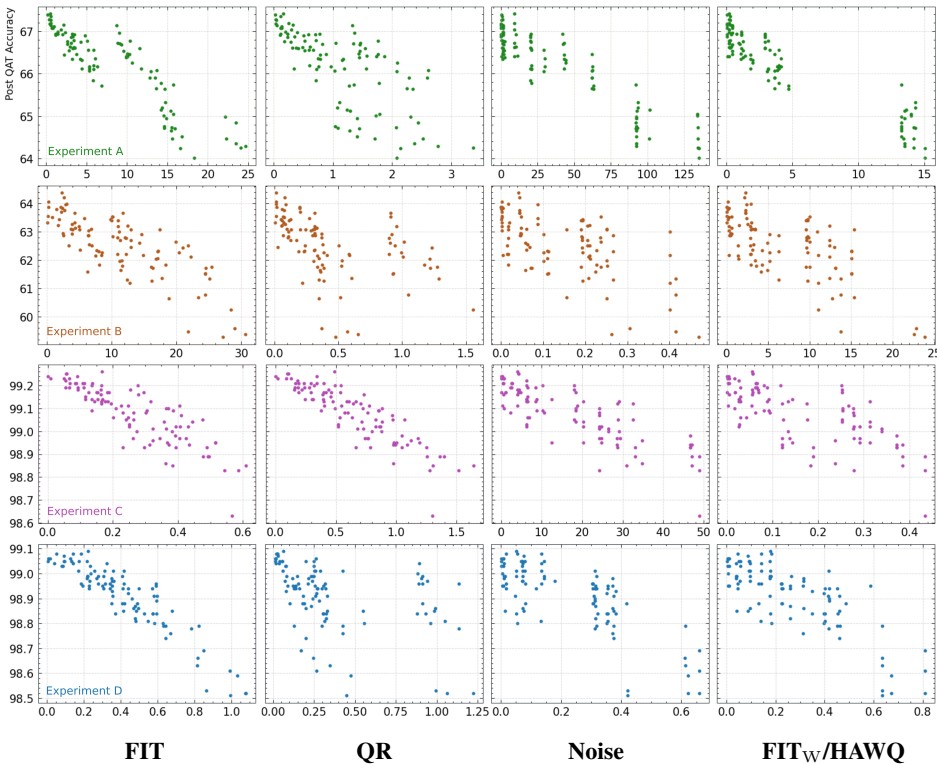

Figure 3: Plots of the chosen predictive heuristic against final model performance.

### 4.3 SEMANTIC SEGMENTATION

In this section, we illustrate the ability of FIT to generalise to larger datasets and architectures, as well as more diverse tasks. We choose to quantify the effects of MPQ on the U-Net architecture (Ronneberger et al., 2015), for the Cityscapes semantic segmentation dataset (Cordts et al., 2016). Additional analysis of BERT on SST-2 is shown in Appendix A. In this experiment, we train 50 models with randomly generated bit configurations for both weights and activations. For evaluation, we use Jaccard similarity, or more commonly, the mean Intersection over Union (mIoU). The final correlation coefficient is between FIT and mIoU. EF trace computation is stopped at a tolerance of 0.01, occurring at 82 individual iterations. The weight and activation traces for the trained, full precision, U-Net architecture is shown in Figure 4. Figure 4(c) shows the correlation between FIT and final model performance for the U-Net architecture on the Cityscapes dataset. In particular, we obtain a high final rank correlation coefficient of **0.86**.

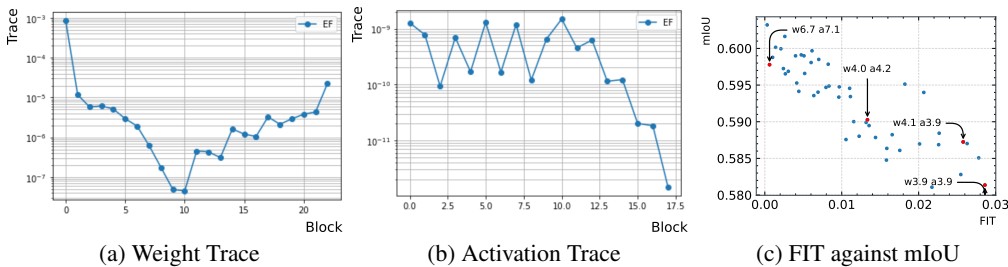

(a) Weight Trace    (b) Activation Trace    (c) FIT against mIoU

Figure 4: EF weight **(a)** and activation **(b)** traces for U-Net architecture on the Cityscapes dataset. **(c)** FIT against mIoU for 50 random MPQ configurations of U-Net on Cityscapes semantic segmentation. Example configurations are highlighted, showing the average bit precision for weights and activations.

### 4.4 FURTHER PRACTICAL DETAILS

**Small perturbations** In Section 3, we assume quantization noise, $\delta\theta$, is small with respect to the parameters themselves. This allows us to reliably consider a second-order approximation of the divergence. In Figure 5, we plot every parameter within the model for every single quantization configuration from experiment A. Almost all parameters adhere to this approximation. We observe that our setting covers most practical situations, and we leave characterising more aggressive quantization (1/2 bit) to future work.

**Distributional shift** Modern DNNs often over-fit to their training dataset. More precisely, models are able to capture the small distributional shifts between training and testing subsets. FIT is computed from the trained model, using samples from the training dataset. As such, the extent to which FIT captures the final quantized model performance on the test dataset is, therefore, dependent on model over-fitting. Consider for example dataset D. We observe a high correlation of **0.98** between FIT and final *training* accuracy, which decreases to **0.90** during *testing*. This is further demonstrated in Figure 5. For practitioners, FIT is more effective where over-fitting is less prevalent.

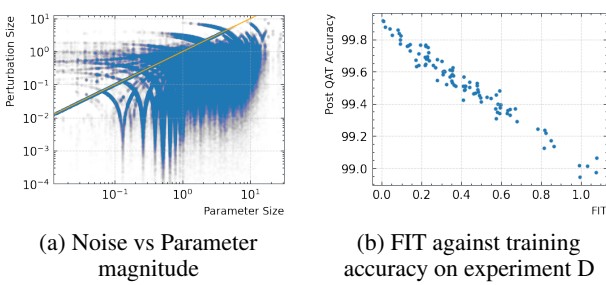

(a) Noise vs Parameter
magnitude

(b) FIT against training
accuracy on experiment D

Figure 5: **(a)** Noise vs parameter magnitude. The line indicates equal magnitude, and is shown for reference. **(b)** FIT against final training accuracy for experiment D. The correlation coefficient, in this case, is 0.98.

## 5 CONCLUSION

In this paper, we introduced FIT, a metric for characterising how quantization affects final model performance. Such a method is vital in determining high-performing mixed-precision quantization configurations which maximise performance given constraints on compression.

We presented FIT from an information geometric perspective, justifying its general application and connection to the Hessian. By applying a quantization-specific noise model, as well as using the empirical fisher, we obtained a well-grounded and practical form of FIT.

Empirically, we show that FIT correlates highly with final model performance, remaining consistent across varying datasets, architectures and tasks. Our ablation studies established the importance of including activations. Moreover, FIT fuses the sensitivity contribution from parameters and activations yielding a single, simple to use, and informative metric. In addition, we show that FIT has very favourable convergence properties, making it orders of magnitude faster to compute. We also explored assumptions which help to guide practitioners.

Finally, by training hundreds of MPQ configurations, we obtained correlation metrics from which to demonstrate the benefits of using FIT. Previous works train a small number of configurations. We encourage future work in this field to replicate our approach, for meaningful evaluation.

**Future Work.**

Our contribution, FIT, worked quickly and effectively in estimating the final performance of a quantized model. However, as noted in Section 4, FIT is susceptible to the distributional shifts associated with model over-fitting. In addition, FIT must be computed from the *trained*, full-precision model. We believe dataset agnostic methods provide a promising direction for future research, where the MPQ configurations can be determined from initialisation.

## REPRODUCIBILITY STATEMENT

We have included sample code for generating the parameter and activation traces, as well as generating and analysing quantized models. Further experimental details are presented in Appendix E. Relevant complete proofs are shown in Appendices F and G.

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

# A  ADDITIONAL EXPERIMENTS: BERT ON SST-2

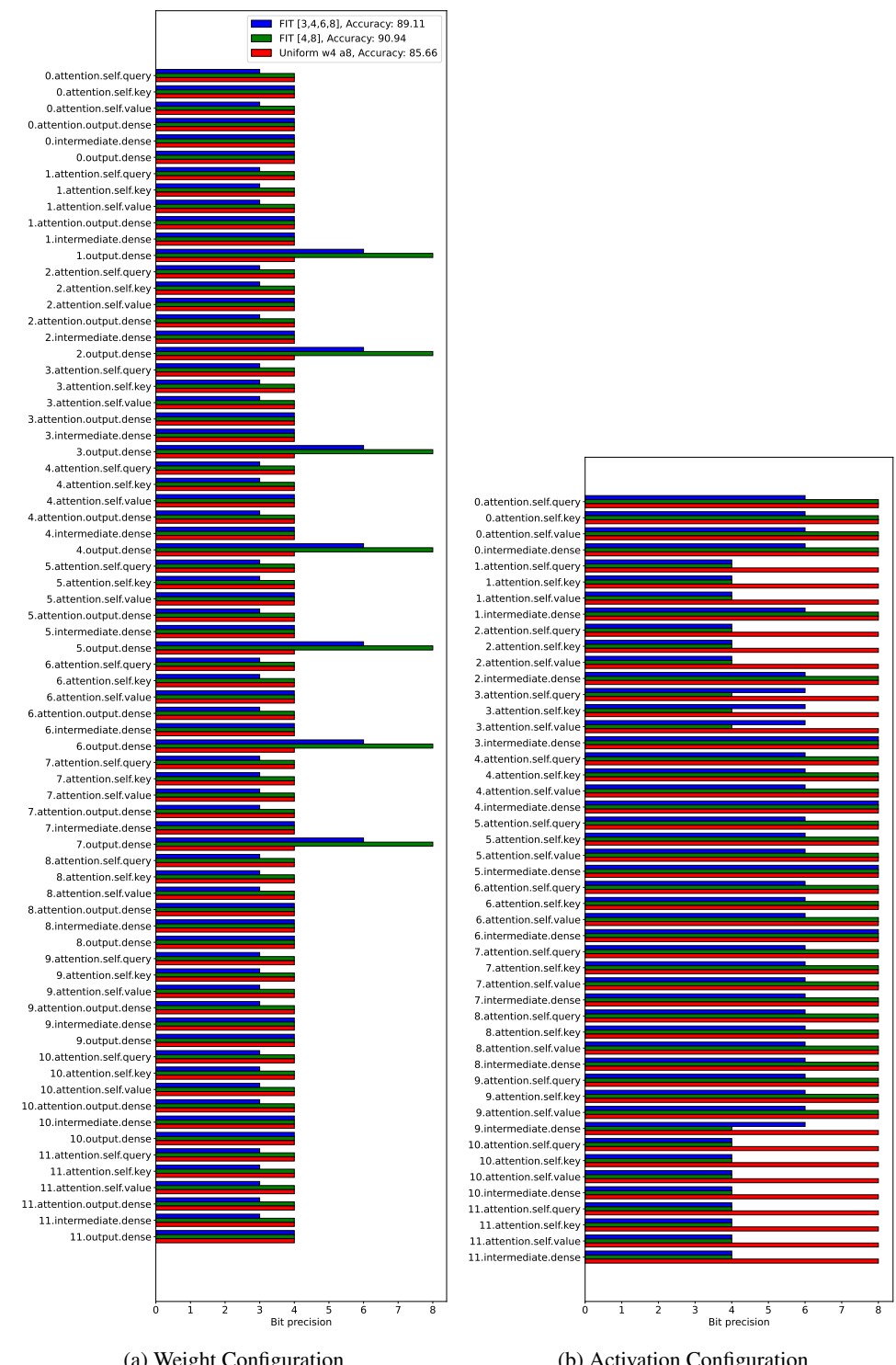

(a) Weight Configuration          (b) Activation Configuration

Figure 6: Comparison between bit precision configurations for the uniform scheme: (W4, A8), and comparable (BOPs) configurations generated via FIT.

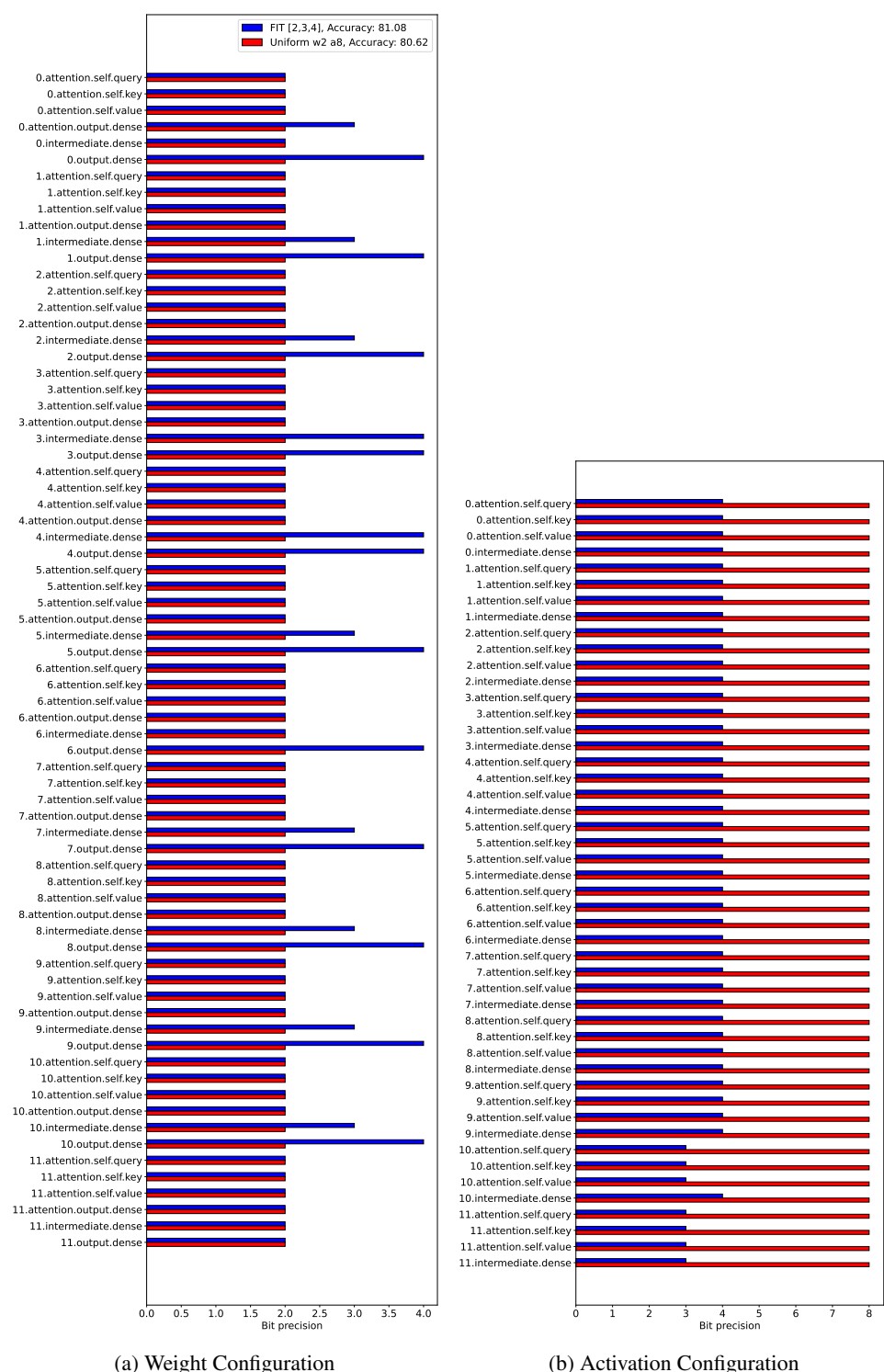

(a) Weight Configuration    (b) Activation Configuration

Figure 7: Comparison between bit precision configurations for the uniform scheme: (W2, A8), and comparable (BOPs) configurations generated via FIT.

This section extends our experimental evaluation of FIT to other challenging datasets and benchmarks. Notably, we focus on quantizing BERT-base Devlin et al. (2018) for the SST-2 dataset Socher et al. (2013). BERT-base is quantized via layer-wise mixed precision simulated asymmetric quantization of

weights and activations. We chose 50 distinct MPQ configurations for BERT on SST-2 across a range of bit precisions [2,3,4,6,8]. We obtain a final correlation score of **0.752** between post-fine-tuning accuracy and FIT score for these MPQ configurations. Similar to other experiments, this demonstrates that FIT has an excellent predictive power of the impact of quantization, even for large and deep models such as BERT. To illustrate a use case for FIT, Figures 6 and 7 show highly non-trivial MPQ configurations compared to the uniform baseline methods constrained by operations (BOPs). In this case, we obtain higher MPQ accuracy at comparable BOPs, showing the superior performance of FIT in selecting high-performing MPQ configurations. Notably, FIT can trade off with any arbitrary hardware constraint (e.g. latency, power), making it very flexible.

From a computational perspective, given a batch size of 256, FIT converged with a variance of 0.73 in very few iterations. Conversely, the Hessian trace method could not properly converge even after 200 iterations, which is reflected in a variance of 5256. This granularity of layer-wise quantization renders Hessian-based methods challenging to use in practice. The EF traces for the weights and activations of BERT are shown in Figure 8.

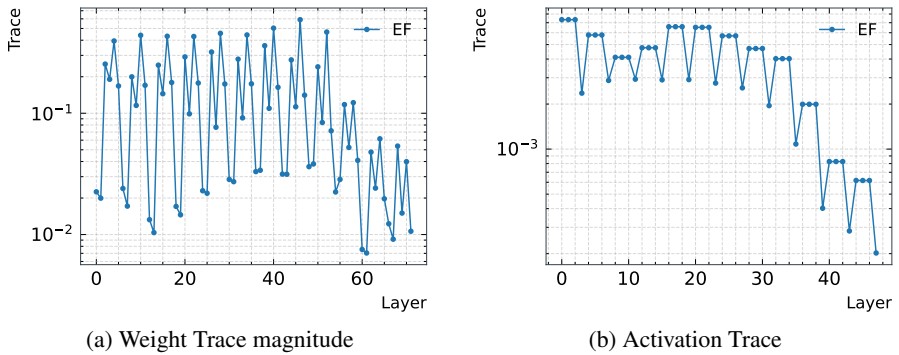

(a) Weight Trace magnitude          (b) Activation Trace

Figure 8: EF weight (**a**) and activation (**b**) traces for BERT-base on the SST-2 dataset.

# B   QUANTIZATION AWARE TRAINING (QAT)

Quantization can lead to significant performance degradation Gholami et al. (2021). It is, therefore, necessary to perform additional quantization aware training (QAT) in order to recover this lost performance. QAT involves simulating the effects of quantization during training. The quantization function $Q(\theta)$ is applied to the floating point parameter values during the forward pass of the network. $Q(\theta)$ is piece-wise flat. As a result, to propagate gradients, we use a straight-through estimator (STE) (Hubara et al., 2016). In effect, this bypasses the quantization function during gradient computation in the backwards pass. Figure 9 illustrates this process. QAT also involves learning the quantization ranges. For weights, a max-min approach is taken, whilst for activations, an exponential moving average is used. As a result, the scaling and zero points for quantization are mapped correctly. In addition, batch-normalisation parameters can be folded into weights for efficiency.

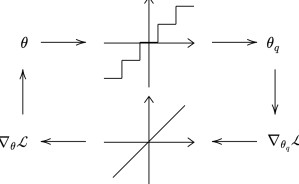

Figure 9: Overview QAT - $Q(\theta)$ is applied during the forward pass, and an STE is used in the backwards pass.

### B.1 FURTHER QUANTIZATION DETAILS

During our experiments, we employ layer-wise symmetric/asymmetric simulated quantization. In addition, layer-wise quantization ranges are accumulated for a short period (30 iterations) at the beginning of QAT to accurately tune the quantization function across the expected range. Note that this is particularly important for activation quantization. As a result of the simulated nature of the quantization, we do not perform batch-norm folding in our experiments. However, this may improve results further, given the additional scaling factor per channel (rather than just per-layer).

Notably, investigating FIT combined with other more complicated quantization schemes such as those proposed by Chang et al. (2021b) and Chang et al. (2021a) provide appealing directions for future work. In particular, we note that due to the formulation of FIT, differing quantization schemes are realised by changing the quantization noise model. We look forward to investigating this in future.

## C ACTIVATION TRACES

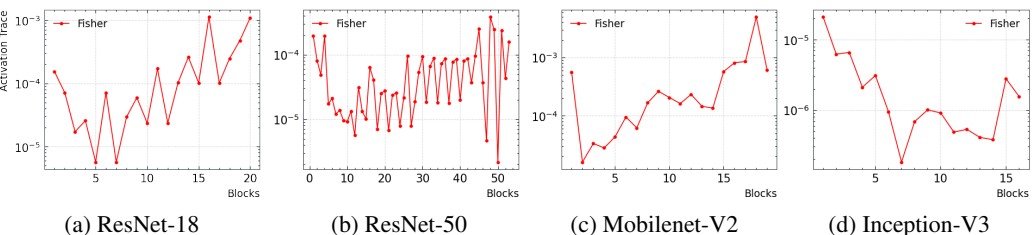

(a) ResNet-18    (b) ResNet-50    (c) Mobilenet-V2    (d) Inception-V3

Figure 10: EF Activation traces for four classification models.

## D ESTIMATOR COMPARISON

For the four classification models considered, Tables 3 and 4 indicate the estimator variances and iteration times associated with the EF and Hessian for a variety of batch sizes. Whilst the EF exhibits expected variance reduction behaviour, the Hessian does not, and requires a minimum, model-dependent, batch size to achieve stable behaviour. In all cases, the EF estimator variance is orders of magnitude lower than that of the Hessian. Means and variances are estimated over 3 runs of 200 samples. Deviations are normalised w.r.t. the trace magnitude, taking the average across blocks/layers. This ensures all the statistics are comparable, regardless of convergence bias. The relative speedup associated with a fixed tolerance is computed as follows:

$$ s = \frac{\sigma_H^2 \cdot t_H}{\sigma_{EF}^2 \cdot t_{EF}} $$

Where $\sigma^2$ indicates the estimator variance and $t$ is iteration time. This follows the Monte-Carlo estimate variance properties.

## E FURTHER EXPERIMENTAL DETAILS

For each of experiments A, B, C and D, we trained 100 convolutional classifiers on the Cifar10 and Mnist datasets. In both cases, we performed experiments with and without the inclusion of batch normalisation layers before each activation. The architecture used consisted of three convolutional layers, followed by a fully connected classification head, with ReLU activations in between. The first two blocks are also followed by MaxPool layers. This is shown in Figure 11. Between the Mnist and Cifar10 datasets, the number of filters was scaled by two. To obtain the data, we first trained a full precision version of the network for 50 epochs using the Adam optimizer. A learning rate of

Table 3: Estimator Variances associated with the EF and Hessian for a variety of batch sizes, over 3 runs of 200 iterations for each model and batch size.

| Batch Size | EF | | | | | Hessian | | | | |
|---|---|---|---|---|---|---|---|---|---|---|
| | 1 | 2 | 3 | Mean | Stdev | 1 | 2 | 3 | Mean | Stdev |
| ResNet-18 | | | | | | | | | | |
| 4 | 0.82 | 0.99 | 1.27 | 1.03 | 0.23 | 10.64 | 9.77 | 6.89 | 9.10 | 1.96 |
| 8 | 0.61 | 0.42 | 0.61 | 0.55 | 0.11 | 5.01 | 4.28 | 4.38 | 4.56 | 0.40 |
| 16 | 0.33 | 0.30 | 0.26 | 0.30 | 0.03 | 1.77 | 2.32 | 2.03 | 2.04 | 0.27 |
| 32 | 0.13 | 0.18 | 0.15 | **0.15** | **0.03** | 1.09 | 1.07 | 1.11 | **1.09** | **0.02** |
| ResNet-50 | | | | | | | | | | |
| 4 | 1.81 | 2.44 | 2.03 | 2.10 | 0.32 | - | - | - | - | - |
| 8 | 1.15 | 1.06 | 1.19 | 1.13 | 0.07 | 22.57 | 45.45 | 76.14 | 48.05 | 26.88 |
| 16 | 0.55 | 0.71 | 0.51 | 0.59 | 0.11 | 17.68 | 22.41 | 14.56 | 18.21 | 3.95 |
| 32 | 0.26 | 0.34 | 0.32 | **0.31** | **0.04** | 8.60 | 5.65 | 6.49 | **6.91** | **1.52** |
| MobileNet-V2 | | | | | | | | | | |
| 4 | 1.18 | 1.29 | 1.03 | 1.17 | 0.13 | - | - | - | - | - |
| 8 | 0.67 | 0.61 | 0.71 | 0.66 | 0.05 | 17.65 | 72.02 | 34.19 | 41.28 | 27.87 |
| 16 | 0.36 | 0.35 | 0.37 | 0.36 | 0.01 | 9.36 | 10.54 | 5,092.34 | 9.95 | 0.84 |
| 32 | 0.25 | 0.24 | 0.22 | **0.24** | **0.01** | 4.47 | 4.75 | 5.23 | **4.81** | **0.38** |
| Inception-V3 | | | | | | | | | | |
| 4 | 3.97 | 3.00 | 2.77 | 3.24 | 0.64 | - | - | - | - | - |
| 8 | 1.92 | 1.24 | 1.95 | 1.70 | 0.40 | - | - | - | - | - |
| 16 | 0.89 | 0.66 | 0.69 | 0.75 | 0.13 | 31.55 | 65.36 | 109.84 | 68.92 | 39.26 |
| 32 | 0.44 | 0.45 | 0.39 | **0.43** | **0.03** | 13.81 | 13.96 | 13.10 | **13.62** | **0.46** |

Table 4: Iteration times associated with the EF and Hessian for a variety of batch sizes, averaged over 3 runs of 200 iterations for each model and batch size.

| Batch Size | EF (ms) | | | | | Hessian (ms) | | | | |
|---|---|---|---|---|---|---|---|---|---|---|
| | 1 | 2 | 3 | Mean | Stdev | 1 | 2 | 3 | Mean | Stdev |
| ResNet-18 | | | | | | | | | | |
| 4 | 11.06 | 11.29 | 11.29 | 11.21 | 0.13 | 41.91 | 40.99 | 40.58 | 41.16 | 0.68 |
| 8 | 16.80 | 16.86 | 16.98 | 16.88 | 0.09 | 62.80 | 61.54 | 61.16 | 61.83 | 0.86 |
| 16 | 24.07 | 24.46 | 24.12 | 24.22 | 0.21 | 100.01 | 95.40 | 98.38 | 97.93 | 2.34 |
| 32 | 47.81 | 47.77 | 47.75 | **47.78** | **0.03** | 186.92 | 186.80 | 185.90 | **186.54** | **0.56** |
| ResNet-50 | | | | | | | | | | |
| 4 | 30.89 | 30.77 | 30.75 | 30.80 | 0.07 | 150.69 | 148.17 | 149.08 | 149.31 | 1.27 |
| 8 | 47.90 | 48.50 | 48.49 | 48.30 | 0.34 | 196.37 | 201.74 | 200.99 | 199.70 | 2.91 |
| 16 | 83.67 | 83.54 | 82.94 | 83.38 | 0.39 | 332.11 | 341.01 | 338.11 | 337.08 | 4.54 |
| 32 | 152.44 | 151.92 | 151.69 | **152.02** | **0.38** | 639.62 | 639.82 | 637.96 | **639.13** | **1.02** |
| MobileNet-V2 | | | | | | | | | | |
| 4 | 19.32 | 19.38 | 19.57 | 19.42 | 0.13 | 709.06 | 705.77 | 704.53 | 706.45 | 2.34 |
| 8 | 21.49 | 21.87 | 21.96 | 21.77 | 0.25 | 913.47 | 917.50 | 914.63 | 915.20 | 2.07 |
| 16 | 33.85 | 33.83 | 33.73 | 33.80 | 0.06 | 1,460.03 | 1,464.95 | 1,460.89 | 1,461.96 | 2.62 |
| 32 | 58.65 | 58.80 | 59.07 | **58.84** | **0.21** | 2,570.02 | 2,575.81 | 2,574.66 | **2,573.50** | **3.06** |
| Inception-V3 | | | | | | | | | | |
| 4 | 56.30 | 59.83 | 57.00 | 57.71 | 1.87 | 265.04 | 276.99 | 279.17 | 273.73 | 7.61 |
| 8 | 83.77 | 83.21 | 82.41 | 83.13 | 0.68 | 325.06 | 335.16 | 332.37 | 330.86 | 5.21 |
| 16 | 135.46 | 132.97 | 132.58 | 133.67 | 1.56 | 530.60 | 537.80 | 535.57 | 534.66 | 3.69 |
| 32 | 236.07 | 235.14 | 235.09 | **235.43** | **0.55** | 901.72 | 908.35 | 955.94 | **905.04** | **4.69** |

0.01 was chosen, and increased to 0.1 with the inclusion of batch normalization. A cosine-annealing learning rate schedule was used. We then used this trained full precision model as a checkpoint to initialise our randomly chosen mixed precision configurations, and training was continued for another 30 epochs with a learning rate reduction of 0.1, using the same schedule. Quantization configurations were chosen uniformly at random from the possible set of bit precisions: [8,6,4,3]. In these cases, initialisation and training were identical across all MPQ models, so as to compare the final performance.

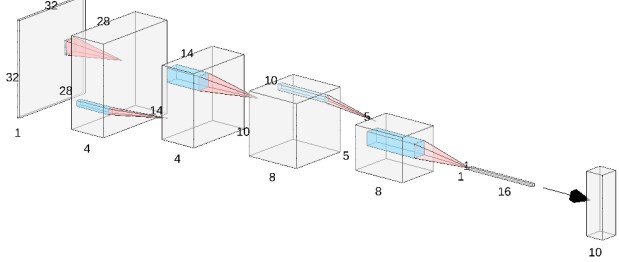

Figure 11: Small convolutional classifier architecture used in the experiments detailed in Section 4

### E.1 FURTHER DETAILS FOR COMPARISON METRICS

**QR:**  The QR baseline represents the use of the quantization ranges to replace the EF as the sensitivity metric:

$$\sum_l^L \frac{1}{|\theta_{max} - \theta_{min}|} \cdot \left[ \frac{\theta_{max} - \theta_{min}}{2^{b_l} - 1} \right]^2$$

**BN:**  the BN baseline represents the use of the batch-norm scaling factor $\gamma$ to replace the EF as the sensitivity metric.

$$\sum_l^L \frac{1}{\gamma} \cdot \left[ \frac{\theta_{max} - \theta_{min}}{2^{b_l} - 1} \right]^2$$

**FIT$_{W/A}$:**  To obtain FIT$_W$, we remove the component which takes into account activation quantization sensitivity. Similarly, to obtain FIT$_A$, we remove the component which takes into account weight quantization sensitivity. Recall that in Section 3, we extend parameter space to include the activation statistics. In these ablations, we remove (W) or retain (A) these contributions.

## F  QUANTIZATION AND NOISE MODEL

The following analysis serves to motivate the direct connection between model perturbation and bit configuration. During quantization, it is common historically to assume the quantization error is uncorrelated with the original signal, yielding an approximately uniform distribution. This leads to an important assumption: The quantization error of each parameter is independent and uniformly distributed with a mean zero. Figure 12 serves to motivate the validity of this assumption.

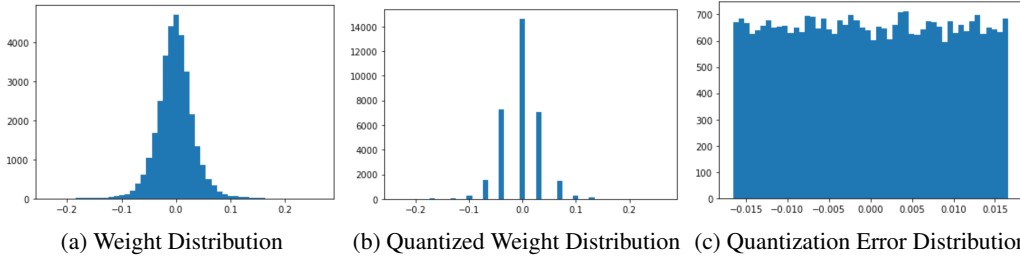

(a) Weight Distribution   (b) Quantized Weight Distribution  (c) Quantization Error Distribution

Figure 12: ResNet-18, block 12, quantization distribution analysis showing the validity of the uniform noise assumption.

Under this assumption, it is possible to evaluate this noise power:

$$\mathbb{E}[\delta\theta^2] = \frac{\Delta^2}{12},$$

where $\Delta$ denotes the step width of quantization, and can be modelled directly by the quantization scheme being used. In this case, uniform (min-max) quantization. Letting $Q(\theta) : \theta \to \theta_q$ denote the quantization function, uniform quantization can be expressed as follows:

$$Q(\theta) = \frac{\theta - \theta_{min}}{\Delta} + \theta_{min}$$

$$\Delta = \frac{\theta_{max} - \theta_{min}}{2^b - 1}$$

where $b$ is the quantization bit precision. This yields the following quantization noise power:

$$\mathbb{E}[\delta\theta^2] = \frac{1}{12}\left[\frac{\theta_{max} - \theta_{min}}{2^b - 1}\right]^2$$

Consider now our revised model perturbation (w.l.o.g. remove the constant factor),

$$\sum_l^L Tr(\hat{I}(\theta_l)) \cdot \left[\frac{\theta_{max} - \theta_{min}}{2^{b_l} - 1}\right]^2.$$

Each bit configuration - $\{b_l\}_1^L$ - yields a unique FIT value from which we can estimate final performance.

## G FIT DETAILS

### G.1 STANDARD RELATIONS IN INFORMATION GEOMETRY

**Proposition 1.** *The EF defines an estimate for the natural metric tensor of the statistical manifold induced by the parameterised model, obtained infinitesimally (in this case) from an expansion of the KL divergence.*

*Proof.* For brevity, denote $p(z, \theta)$ as $p_\theta$. Here we describe the discrete case, however, the continuous follows similarly.

$$D_{KL}(p_\theta||p_{\theta+\delta\theta}) = \sum_D p_\theta \log \frac{p_\theta}{p_{\theta+\delta\theta}}$$

$$\approx \sum_D p_\theta \log p_\theta - p_\theta \left[\log p_\theta + \frac{\nabla_\theta p_\theta}{p_\theta}^T \delta\theta + \frac{1}{2}\delta\theta^T \left(\frac{\nabla_\theta^2 p_\theta}{p_\theta} - \frac{\nabla_\theta p_\theta \nabla_\theta p_\theta^T}{p_\theta^2}\right)\delta\theta\right]$$

$$\approx -\left[\sum_D \nabla_\theta p_\theta\right]^T \delta\theta - \frac{1}{2}\delta\theta^T \left[\sum_D \nabla_\theta^2 p_\theta\right]\delta\theta + \frac{1}{2}\delta\theta^T \left[\sum_D p_\theta \nabla_\theta \log p_\theta \nabla_\theta \log p_\theta^T\right]\delta\theta$$

$$\approx -\left[\nabla_\theta \sum_D p_\theta\right]^T \delta\theta - \frac{1}{2}\delta\theta^T \left[\nabla_\theta^2 \sum_D p_\theta\right]\delta\theta + \frac{1}{2}\delta\theta^T \left[\sum_D p_\theta \nabla_\theta \log p_\theta \nabla_\theta \log p_\theta^T\right]\delta\theta$$

$$\approx \frac{1}{2}\delta\theta^T \left[\sum_D p_\theta \nabla_\theta \log p_\theta \nabla_\theta \log p_\theta^T\right]\delta\theta$$

$$\approx \frac{1}{2}\delta\theta^T \left[\frac{1}{N} \sum_D \nabla_\theta \log p_\theta \nabla_\theta \log p_\theta^T\right]\delta\theta$$

$$= \frac{1}{2}\delta\theta^T \hat{I}(\theta)\delta\theta$$

$\square$

## G.2 Obtaining FIT

**Proposition 2.** *FIT is obtained (under mild noise assumptions) by taking expectation over the Fisher-Rao metric.*

*Proof.*

$$\Omega = \mathbb{E}\left[\delta\theta^T I(\theta)\delta\theta\right]$$
$$= \mathbb{E}[\delta\theta]^T I(\theta)\mathbb{E}[\delta\theta] + Tr(I(\theta)Cov[\delta\theta])$$

We assume that the random noise is symmetrically distributed around mean of zero: $\mathbb{E}[\delta\theta] = 0$, and uncorrelated: $Cov[\delta\theta] = Diag(\mathbb{E}[\delta\theta^2])$. This yields the FIT heuristic:

$$\Omega = Tr\left(I(\theta)Diag(\mathbb{E}[\delta\theta^2])\right)$$

And in its block-wise empirical form, with an approximate noise model:

$$\Omega = \sum_l^L Tr(\hat{I}(\theta_l)) \cdot Diag(\hat{\mathbb{E}}_l[\delta\theta^2]).$$

$\square$

In cases where we compute a numerical approximation of our noise model: $\frac{1}{n(l)} \cdot ||\delta\theta||_l^2$, rather than averaging over many trained models, we leverage the high dimensionality of the parameters in each layer to obtain a good approximation.

## G.3 Connections to the Hessian

**Proposition 3.** *The Fisher Information Matrix is equal to the expectation of the Hessian under the correctly specified model.*

*Proof.*

$$\mathbb{E}_{p_{\theta^*}(z)}\left[\nabla_\theta^2 f(z,\theta)\right] = \mathbb{E}_{p_{\theta^*}(x,y)}\left[-\frac{\nabla_\theta^2 p(y|x,\theta)}{p(y|x,\theta)}\right] + \mathbb{E}_{p_{\theta^*}(x,y)}\left[\frac{\nabla_\theta p(y|x,\theta)\nabla_\theta p(y|x,\theta)^T}{p(y|x,\theta)^2}\right] \quad (1)$$

Taking the first term of the right-hand side (RHS). We assume in this case the function is sufficiently smooth such that the order of integration and (second) differentiation can be exchanged. In this case, the first term on the RHS evaluates to 0:

$$\mathbb{E}_{p_{\theta^*}(x,y)}\left[-\frac{\nabla_\theta^2 p(y|x,\theta)}{p(y|x,\theta)}\right] = -\int_{\mathcal{Z}} \frac{\nabla_\theta^2 p(y|x,\theta)}{p(y|x,\theta)} p(y|x,\theta)dz \quad (2)$$

$$= -\int_{\mathcal{Z}} \nabla_\theta^2 p(y|x,\theta)dz \quad (3)$$

$$= -\nabla_\theta^2 \int_{\mathcal{Z}} p(y|x,\theta)dz \quad (4)$$

$$= -\nabla_\theta^2[1] = 0 \quad (5)$$

$$(6)$$

The second term on the RHS is then simplified as follows, yielding our result:

$$\mathbb{E}_{p_{\theta^*}(z)}\left[\nabla_\theta^2 f(z,\theta)\right] = \mathbb{E}_{p_{\theta^*}(x,y)}[\nabla_\theta \log p(y|x,\theta)\nabla_\theta \log p(y|x,\theta)^T] = I(\theta) \quad (7)$$

$\square$

**Proposition 4.** *Provided $\hat{\theta}_n$ is a consistent estimator for the true model parameters $\theta^*$, the Empirical Fisher Information will converge in probability to the Fisher Information as $n \to \infty$.*

*Proof.*

$$\hat{I}(\hat{\theta}_n) - I(\theta^*) = [\hat{I}(\hat{\theta}_n) - I(\hat{\theta}_n)] + [I(\hat{\theta}_n) - I(\theta^*)] \tag{8}$$

Considering the first term on the RHS, we are able to apply the uniform law of large numbers,

$$[\hat{I}(\hat{\theta}_n) - I(\hat{\theta}_n)] = \frac{1}{N} \sum_{i=1}^{N} \nabla f(z_i, \hat{\theta}_n) \nabla f(z_i, \hat{\theta}_n)^T - \mathbb{E}_{p_{\theta^*}(x,y)}[\nabla_\theta f(z, \hat{\theta}_n) \nabla_\theta f(z, \hat{\theta}_n)^T] \tag{9}$$

$$\leq \sup_{\theta \in \Theta} \left| \frac{1}{N} \sum_{i=1}^{N} \nabla f(z_i, \theta) \nabla f(z_i, \theta)^T - \mathbb{E}_{p_{\theta^*}(x,y)}[\nabla_\theta f(z, \theta) \nabla_\theta f(z, \theta)^T] \right| \tag{10}$$

$$\xrightarrow{p_{\theta^*}} 0, \tag{11}$$

provided the following regularity conditions are upheld:

1. $\Theta$ must be compact

2. $f(z, \theta)$ continuous at each $\theta \in \Theta$ for almost all $z \in \mathcal{Z}$

3. The aforementioned dominating function must be finite

Now considering the second term on the RHS. Our estimator in this case is defined as consistent, and therefore $\hat{\theta}_n \xrightarrow{p_{\theta^*}} \theta^*$ as $n \to \infty$. Via the continuous mapping theorem, $[I(\hat{\theta}_n) - I(\theta^*)] \xrightarrow{p_{\theta^*}} 0$. We therefore arrive at the desired result. $\square$

### G.4 COMPUTATION

**Proposition 5.** *The trace of the EF can be computed via the sum of the square of the gradient.*

*Proof.* The trace is a linear operator, which allows each individual estimator to be extracted from the summation. Additionally, the trace of the second moment matrix can then be computed by the norm of the vector of parameters from which it is composed:

$$\begin{aligned} Tr[\hat{I}(\theta)] &= Tr\left[ \frac{1}{N} \sum_{i=1}^{N} \nabla f(z_i, \theta) \nabla f(z_i, \theta)^T \right] \\ &= \frac{1}{N} \sum_{i=1}^{N} Tr\left[ \nabla f(z_i, \theta) \nabla f(z_i, \theta)^T \right] \\ &= \frac{1}{N} \sum_{i=1}^{N} \nabla f(z_i, \theta)^T \nabla f(z_i, \theta) \\ &= \frac{1}{N} \sum_{i=1}^{N} ||\nabla f(z_i, \theta)||^2 \end{aligned}$$

$\square$

**Proposition 6.** *The variance of the Hutchinson estimator is given by:* $\mathbb{V}[r_i^T H r_i] = 2\left( ||H||_F^2 - \sum_i H_{ii}^2 \right)$

*Proof.* Assuming $r$ follows a Rademacher distribution, the first four moments are as follows: $m = (0, 1, 0, 1)$. As such, we can compute the variance of the quadratic form:

$$\mathbb{V}[r_i^T H r_i] = \left(m_4 - 3m_2^2\right) \cdot \sum_i H_{ii}^2 + m_2^2 \cdot \left(Tr(H)^2 + 2 \cdot Tr(H^2)\right) - \mathbb{E}[r_i^T H r_i]^2$$

$$= 2\left(||H||^2 - \sum_i H_{ii}^2\right)$$

$\square$

## H    ENVIRONMENTAL ANALYSIS

During the course of this research, we estimate our total emissions to be 10.8 kgCO$_2$eq. $O(100)$ hours of computation was performed on an RTX 2080 Ti (TDP of 250W), using a private infrastructure which has a carbon efficiency of 0.432 kgCO$_2$eq/kWh. Estimations were conducted using the MachineLearning Impact calculator presented by Lacoste et al. (2019).

