# OpenReview forum: "FIT: A Metric for Model Sensitivity"
_ICLR.cc/2023/Conference — ICLR 2023 poster_

### Official Review · Reviewer_cnF4 · 2022-10-25

**Confidence:** 2
**Correctness:** 3
**Technical Novelty And Significance:** 3
**Empirical Novelty And Significance:** 4
**Recommendation:** 8

**Clarity, Quality, Novelty And Reproducibility:**

Quality
The work provides strong intuition of the importance of evaluating the quality of a network post quantization without having to retrain the network. In terms of general concepts and building new ideas from foundational ones, the quality of the work is very good.

Clarity
The well-laid out context as well as thorough explanation of the method itself lend a lot of clarity to the reader.

Originality
While the general ideas are firmly established in existing work, the novelty of the proposed work comes from combining them and simplifying them for the task at hand.

**Strength And Weaknesses:**

Strengths
- The proposed work provides a clear and concise explanation of the problem domain being tackled. This guides the reader from the general impact of the proposed work towards the specifics of the problem being handled.
- The evaluation schemes provided for the estimator as well as the application on various networks provide a good foundation.

Weaknesses
- Could the authors provide citations for the statement: "different layers, within different architectures, respond differently to quantization". (Pg. 1, Introduction, Paragraph 4, Line 1)
- The proposal of rank-correlation as a novel evaluation criterion is mentioned as a core contribution towards the end of the introduction. Could the authors describe possible evaluation schemes to compare against prior works, as baselines to judge the proposed scheme. Currently, rank-correlation exists as a readily available tool that has been repurposed for evaluating the task at hand. Multiple baselines could help highlight the contribution.
- Could the authors provide citations for the statement: " which is insufficient for prediction, and more challenging for practitioners to implement. In addition, trace computation can become very expensive for large networks". (Pg. 3, Paragraph 1, Lines 9-11).
- While the evaluation of the EF estimator in Table 1 shows a relatively model-agnostic behavior, it would be instructive to include evaluations with alternative DNN architectures that branch away from bottleneck-like structures (more densely packed connections) and alternative datasets, to help firmly establish the agnostic behavior observed.
- Could the authors normalize the X and Y scales across both plots in Figure 2? The visual comparison would provide a more compelling evidence in that regard.
- Could the authors discuss the type of information provided by FIT_A and FIT_W and try to provide more justification as to how they provide complementary information?

**Summary Of The Paper:**

Model compression via quantization of weights and activations helps improve the efficiency and reduce memory requirements of DNNs.
However, quantifying the impact of quantization across the different layers in a DNN is challenging. For this purpose, the proposed work establishes FIT as a measure of DNN performance, when using multi-precision quantization (MPQ), without the need to retrain the network.
FIT is a metric that combines the impact of quantization on weights as well as activations in a unified manner such that the computation of the metric itself is quick and readily usable.
Thus, FIT can be used to understand and readily gauge the impact of MPQ and avoid the pitfalls of bad performance.

**Summary Of The Review:**

The proposed work establishes the need for a metric to analyze the sensitivity of networks to quantization without having to retrain the network. For this purpose Fisher Information Trace is selected as the measure of choice and its estimation process is defined in the proposed work.
Overall, the evaluation of FIT against the hessian, proving its equivalence for the task at hand, and subsequent evaluation across various application domains helps validate the claims made about FIT.

---

> ### Author Response · Authors · 2022-11-11
> **Response to Reviewer cnF4**
>
> Dear Reviewer
>
> Thank you for your thorough review of our work and suggestions for improvement. We are pleased that you find the paper clear, and that our evaluations provide a solid foundation.
>
> **Rank Correlation for Evaluating MPQ metrics**
>
> It is common for quantization papers to use small numbers of trained models to show the efficacy of their methods. However, given that applications require a broad range of constraints, this granularity of analysis was neither as convincing nor as statistically robust as evaluating across a more significant number of models. Unfortunately, to the best of our knowledge, no previous papers consider this many quantization configurations to facilitate a statistically significant correlation result. This is one of the strengths of our paper and is more convincing than providing a small set of cherry-picked configurations which adhere to arbitrary computational constraints. Whilst this evaluation method is more time-consuming, it is also far more rigorous and demonstrative of generalisability.
>
> **Model Agnostic Variance:**
>
> During this rebuttal period, we extended our empirical evaluation to include BERT-base on SST-2. Not only do we achieve a high-rank correlation of 0.756 across a wide range of configurations (50), but the method also demonstrates the computational improvements associated with the FIT metric (pertaining to your comment). In particular, for a batch size of 256, FIT obtained a variance of 0.73, converging in very few iterations. Conversely, the Hessian method could not converge properly, even after 200 iterations, resulting in a variance of 5256. For layer-wise quantization, this renders Hessian-based methods far more challenging to use. Using larger batch sizes was computationally infeasible. This discussion will be added to the appendix of the rebuttal revision rather than Figure 1 and Table 1.
>
> **Complimentary Information:**
>
> Disentangling the contributions from weight and activation quantization is challenging, as they are inherently intertwined in a layer-wise feedforward neural network. However, although they are implicitly connected via gradient computation, FIT_A and FIT_W compute FIT with respect to the activation and weight quantization configuration, respectively. In this way, the expressions themselves cover disjoint components of the overall quantization configuration. We observe that the performance of FIT improves through a combination of FIT_A and FIT_W. We expect that because we take into account every dimension of compression space ((w1, w2, … , wn), (a1, a2, …, an-1)).
>
> **Additional Changes:** The remaining suggestions for improvement will be included in the rebuttal revision.
>
> Once again, we thank the reviewer for their high score and insightful comments. Based on your detailed comments, you have understood our contribution well. With the above changes and discussion, we hope to alleviate any of your remaining concerns so that you can increase your confidence score.
>
> Best regards,
>
> The Authors

---

### Official Review · Reviewer_uWSp · 2022-10-25

**Confidence:** 4
**Correctness:** 4
**Technical Novelty And Significance:** 4
**Empirical Novelty And Significance:** 3
**Recommendation:** 8

**Clarity, Quality, Novelty And Reproducibility:**

The work is interesting. It proposes to use Fisher information for guiding mixed precision quantization, what to the best of my knowledge hasn't been used for such purpose before. Quality of the submission is good, the work flows logically. Additional details provided in appendix clearly explain all claims and can be useful for reproducing results.

**Strength And Weaknesses:**

The work is clear and well structured. It also flows logically, with a detailed background, good comparison with the previous work, clearly explained FIM metric and experiments that include in-depth comparison with Hessian and analysis of model sensitivity based on the proposed criterion. The method is evaluated on a broad set of models and datasets to evaluate its performance and usability for different tasks.

In general, I don't see any major omissions. Some sections that could be improved to further improve the quality of submissions are:
1) Could you elaborate more on the ideas of determining MPQ configurations from initialization instead of the trained models?
2) Could you provide more details. about the applied quantization scheme? Symmetric, asymmetric, per tensor, per channel? Have you performed experiments with other quantization schemes?
3) The abstract mentioned that hundreds of quantization configurations were used, but there was no clear summary of used configurations in the description, could you please explain it in more detail?

**Summary Of The Paper:**

The presented work proposes a method for estimating model performance and mixed precision configuration without network retraining. The introduced method is based on the Fisher information, what allows for faster computations compared to other existing methods based on e.g., Hessians.

**Summary Of The Review:**

This is a good quality interesting paper with detailed experiments and clear conclusions.

---

> ### Author Response · Authors · 2022-11-11
> **Response to Reviewer  uWSp**
>
> Dear Reviewer
>
> Thank you for your thorough review of our work and suggestions for improvement. We are delighted that you find the paper well-structured, and our conclusions well justified.
>
> **Additional Quantisation Details:**
>
> During our experiments, we employ layer-wise symmetric/asymmetric simulated quantisation. In addition, layer-wise quantisation ranges are accumulated for a short period (30 iterations) at the beginning of QAT to tune the quantisation function across the expected range accurately. Note that this is particularly important for activation quantisation. As a result of the simulated nature of the quantisation, we do not perform batch-norm folding in our experiments. However, this may improve results further, given the additional scaling factor per channel (rather than just per-layer). We will add this discussion to the rebuttal revision appendix. With regards to other quantization schemes, there are many approaches for performing MPQ, and we covered a sample in the literature review. In the paper, we restrict our baseline to layer-wise MPQ, commonly done in practice (https://www.nature.com/articles/s42256-021-00356-5). Investigating FIT combined with other more complicated quantization schemes, such as those proposed by reviewer orGP (cite), provide appealing directions for future work. In particular, we note that due to the formulation of FIT, differing quantization schemes are realised by changing the quantization noise model. We look forward to investigating this in future.
>
> **Further Experimental Details:**
>
> As noted in the abstract, we trained 100’s quantization configurations to compute correlation results. Consequently, it is not practical to list all of them: a total of 500 individual quantization configurations and accuracy values. We feel it would not improve the clarity of the paper. Instead, we highlight configuration examples for various accuracies and FIT values in Figures 3 and 4. We will include this change in the rebuttal revision.
>
> To alleviate any concerns that FIT is not capable of being used to generate non-trivial configurations, consider the following example for BERT on SST-2 (an additional experiment across 50 models provided during this rebuttal period).
>
> *First Baseline Model:*
> - Weights: 4 uniform
> - Activations: 8 uniform
> - Accuracy: 85.66
>
> *Comparable FIT Model [3,4,6,8]*
> - Weights:  [3, 4, 3, 4, 4, 4, 3, 3, 3, 4, 4, 6, 3, 3, 4, 3, 4, 6, 3, 3, 3, 4, 4, 6, 3, 3, 4, 3, 4, 6, 3, 3, 4, 3, 4, 6, 3, 3, 4, 3, 4, 6, 3, 3, 3, 3, 4, 6, 3, 3, 3, 4, 4, 4, 3, 3, 3, 3, 4, 4, 3, 3, 3, 3, 4, 4, 3, 3, 3, 3, 3, 4]
> - Activations: [6, 6, 6, 6, 4, 4, 4, 6, 4, 4, 4, 6, 6, 6, 6, 8, 6, 6, 6, 8, 6, 6, 6, 8, 6, 6, 6, 8, 6, 6, 6, 6, 6, 6, 6, 6, 6, 6, 6, 6, 4, 4, 4, 4, 4, 4, 4, 4]
> - Accuracy: **89.11**
>
> *Comparable FIT Model [4,8]*
> - Weights:  [4, 4, 4, 4, 4, 4, 4, 4, 4, 4, 4, 8, 4, 4, 4, 4, 4, 8, 4, 4, 4, 4, 4, 8, 4, 4, 4, 4, 4, 8, 4, 4, 4, 4, 4, 8, 4, 4, 4, 4, 4, 8, 4, 4, 4, 4, 4, 8, 4, 4, 4, 4, 4, 4, 4, 4, 4, 4, 4, 4, 4, 4, 4, 4, 4, 4, 4, 4, 4, 4, 4, 4]
> - Activations: [8, 8, 8, 8, 4, 4, 4, 8, 4, 4, 4, 8, 4, 4, 4, 8, 8, 8, 8, 8, 8, 8, 8, 8, 8, 8, 8, 8, 8, 8, 8, 8, 8, 8, 8, 8, 8, 8, 8, 4, 4, 4, 4, 4, 4, 4, 4, 4]
> - Accuracy: **90.94**
>
> *Second Baseline Model:*
> - Weights: 2 uniform
> - Activations: 8 uniform
> - Accuracy: 80.62
>
> *Comparable FIT Model [2,3,4]*
> - Weights: [2, 2, 2, 3, 2, 4, 2, 2, 2, 2, 3, 4, 2, 2, 2, 2, 3, 4, 2, 2, 2, 2, 4, 4, 2, 2, 2, 2, 4, 4, 2, 2, 2, 2, 3, 4, 2, 2, 2, 2, 2, 4, 2, 2, 2, 2, 3, 4, 2, 2, 2, 2, 3, 4, 2, 2, 2, 2, 3, 4, 2, 2, 2, 2, 3, 4, 2, 2, 2, 2, 2, 2]
> - Activations: [4, 4, 4, 4, 4, 4, 4, 4, 4, 4, 4, 4, 4, 4, 4, 4, 4, 4, 4, 4, 4, 4, 4, 4, 4, 4, 4, 4, 4, 4, 4, 4, 4, 4, 4, 4, 4, 4, 4, 4, 3, 3, 3, 4, 3, 3, 3, 3]
> - Accuracy: **81.08**
>
> **Quantization from Initialisation:**
>
> Determining optimal MPQ configuration with the FIT metric before any training has taken place remains an open question. We have some foothold on this topic currently. Although individual parameter contributions change during training (with a large subset even collapsing, i.e. lottery tickets), the layer-wise statistics remain (relatively) consistent, possibly an artefact of modern initialisation schemes (He et al.). As a result, it could be possible to perform QAT directly without the need to fine-tune from a complete precision checkpoint. However, the type of initialisation also matters, e.g. https://arxiv.org/pdf/1510.00149.pdf. We are investigating this concurrently, along with the unification of quantization and pruning under the FIT metric. We could discuss the topic further if the reviewer is interested! We planned it as a follow-up contribution.
>
> With the above changes and discussion, we hope to alleviate any of your remaining concerns and take steps to answer questions for future work. Once again, we thank the reviewer for their high score and insightful comments.
>
> Best regards,
>
> The Authors

---

### Official Review · Reviewer_QyLp · 2022-10-26

**Confidence:** 5
**Correctness:** 2
**Technical Novelty And Significance:** 1
**Empirical Novelty And Significance:** 2
**Recommendation:** 3

**Clarity, Quality, Novelty And Reproducibility:**

* Some things are quite unclear on the implementation details. How exactly is the quantization done? Do we fold batch-norm? Do we do asymmetric/symmetric quantization? Per-channel or not? What bit-widths are included for the mixed precision results? I would like to see more details on this in the paper
* The quality of the paper is high, very well polished
* The novelty is not there, as I mentioned in the strenghts/weaknesses above, since literally this metric has been used before in the same context of quantization
* The authors noted they included sample code, so these specific experiments can be reproduced.

**Strength And Weaknesses:**

Strengths:
* It is very clear the authors know what they are talking about, as they mentioned all the relevant literature, and understand the underlying methods/ideas very well.
* An excellent overview of the literature, and the derivations leading to the final FIM objective. This is a great overview for anyone to read that wants to dive into this topic.

Weaknesses
* The metric itself has been introduced before, for both compression and quantization. I have seen it several times in the past. The first mention of it I know in the setting of quantization and compression is from this paper: https://arxiv.org/pdf/1810.06401.pdf - where it is derived from an information theoretical context. There is no novelty in the paper, since it is a small tweak to HAWQ, and the tweak itself already existed.
* I think the paper is lacking in terms of experimental results. It is good the authors included the accuracy comparison w.r.t. the target loss for this method and 'competitive' methods. However, these numbers don't mean much for the performance in the mixed-precision setting. For the correlations in Table 2, I wonder if those are high enough to do a proper mixed-precision setting that is non-trivial. For these networks, how well does the mixed precision method rank the networks, and how well can it find a non-trivial solution? Why is this analysis that was done for semantic segmentation also not done for these other models? I don't like hammering on more results, but since the metric itself is not novel, the onus is on the authors to really show that in this setting this metric works well and is the best we can do. The paper turns into a survey paper, rather than a novel method paper.
* I also wonder about the sense of these metrics in general, and if there is no easier solution. I think this is related to a key underlying assumption at the start of the paper: "The exact (per parameter) perturbations δθ associated with quantization are often unknown."
This is a weird statement. If you 8-bit quantize a layer’s weights, with a given min-max parameter setting, you know exactly what δθ is. There is nothing inherently stochastic about it. Instead of making this stochastic assumption, you could just quantize each layer to the intended potential bit-widths, and measure this on your validation set. Here is your loss function approximation, no surrogate measure needed. This also holds for the activations, where there is arguable a distribution over your data (but not over noise on your model, as the formulation now has it). Please compare what your metric would improve over a simple eval step on a batch of data for per-layer/per-quantizer bit-width settings. In practice, one can generally not set intermediate bit-widths anyway, and generally you'd pick 4, 8, 16 or something, which will likely be almost as efficient as this method.

**Summary Of The Paper:**

The paper finds a more efficient alternative than the Hessian Trace metric used in the mixed-precision HAWQ paper, named FIT, which is a further simplification of said metric based on the empirical fisher. They show the metric to be equivalent in many scenarios compared to the Hessian trace metric.

**Summary Of The Review:**

The paper is extremely well-written, and great in terms of clarity and exhaustiveness on the literature regarding Hessian-like estimates for deep learning.
The method of the paper has been published before, in the context of quantization. The results do convince me that this metric works just as well as the HAWQ metric; They also convince me that they are better than the other metrics the authors found in the literature. However, the authors have not convinced me that these metrics are useful in mixed-precision, that the resulting algorithm actually does something that is non-trivial, and that this is a great way to do mixed precision in general.

I would like to see the following things:
* A proper set of mixed precision results on more than 1 network, not only the semantic segmentation one. The authors included a few more networks for their comparative analysis, but then do not show actual mixed precision results on these nets. This should include showing that the loss approximation is good enough to find non-trivial mixed-precision results
* A comparison to a very simple baseline - Just quantize each layer individually with your intended bit-width, and measure the output loss on a bit of data (few batches e.g.); Your metric is gauranteed to be a less accurate metric for mixed-precision by construction. How well does it perform in practice, and why would your metric be expected to perform better?
* A more thorough experimental setup description. There's a lot of choices to be made in quantization, and these choices matter.

If these are addressed in a satisfactory way, I am very willing to increase my score.

---

> ### Author Response · Authors · 2022-11-11
> **Response to Reviewer QyLp (Part 1)**
>
> Dear Reviewer.
>
> We thank you for your thorough review of our work and suggestions for improvement. We are delighted that you found the paper well-written, and we are flattered that you would recommend it to anyone who wants to dive into the MPQ topic to read it.
>
> We want to alleviate your concerns regarding the lack of novelty and experimental results.
>
> **Additional Experiments**
>
> We concur with your recommendation to expand evaluation results. In particular, we analysed BERT on the SST-2 dataset. The rebuttal revision will include changes to the experimental setup, results and discussion sections (the main body and the appendix). For brevity, we summarise this in our response:
> - *Experimental setup:* we chose 50 distinct MPQ configurations for BERT on SST-2 across a range of bit precisions [2,3,4,6,8]. BERT-base is quantised via layer-wise mixed precision simulated asymmetric quantisation of weights and activations.
> - *Results:* We obtain a final correlation score of 0.752 between post-fine-tuning accuracy and FIT score for these MPQ configurations. Similar to other experiments, this demonstrates that FIT has an excellent predictive power of the impact of quantisation, even for large and deep models such as BERT.
> - To demonstrate a use case for FIT, we list below three different MPQ configurations compared to baseline methods constrained by operations (BOPS), showing the superior performance of FIT in selecting high-performing MPQ configurations. Notably, FIT can trade off with any arbitrary hardware constraint (e.g. latency, power), making it very flexible.
>
> *First Baseline Model:*
> - Weights: 4 uniform
> - Activations: 8 uniform
> - Accuracy: 85.66
>
> *Comparable FIT Model [3,4,6,8]*
> - Weights:  [3, 4, 3, 4, 4, 4, 3, 3, 3, 4, 4, 6, 3, 3, 4, 3, 4, 6, 3, 3, 3, 4, 4, 6, 3, 3, 4, 3, 4, 6, 3, 3, 4, 3, 4, 6, 3, 3, 4, 3, 4, 6, 3, 3, 3, 3, 4, 6, 3, 3, 3, 4, 4, 4, 3, 3, 3, 3, 4, 4, 3, 3, 3, 3, 4, 4, 3, 3, 3, 3, 3, 4]
> - Activations: [6, 6, 6, 6, 4, 4, 4, 6, 4, 4, 4, 6, 6, 6, 6, 8, 6, 6, 6, 8, 6, 6, 6, 8, 6, 6, 6, 8, 6, 6, 6, 6, 6, 6, 6, 6, 6, 6, 6, 6, 4, 4, 4, 4, 4, 4, 4, 4]
> - Accuracy: **89.11**
>
> *Comparable FIT Model [4,8]*
> - Weights:  [4, 4, 4, 4, 4, 4, 4, 4, 4, 4, 4, 8, 4, 4, 4, 4, 4, 8, 4, 4, 4, 4, 4, 8, 4, 4, 4, 4, 4, 8, 4, 4, 4, 4, 4, 8, 4, 4, 4, 4, 4, 8, 4, 4, 4, 4, 4, 8, 4, 4, 4, 4, 4, 4, 4, 4, 4, 4, 4, 4, 4, 4, 4, 4, 4, 4, 4, 4, 4, 4, 4, 4]
> - Activations: [8, 8, 8, 8, 4, 4, 4, 8, 4, 4, 4, 8, 4, 4, 4, 8, 8, 8, 8, 8, 8, 8, 8, 8, 8, 8, 8, 8, 8, 8, 8, 8, 8, 8, 8, 8, 8, 8, 8, 4, 4, 4, 4, 4, 4, 4, 4, 4]
> - Accuracy: **90.94**
>
> *Second Baseline Model:*
> - Weights: 2 uniform
> - Activations: 8 uniform
> - Accuracy: 80.62
>
> *Comparable FIT Model [2,3,4]*
> - Weights: [2, 2, 2, 3, 2, 4, 2, 2, 2, 2, 3, 4, 2, 2, 2, 2, 3, 4, 2, 2, 2, 2, 4, 4, 2, 2, 2, 2, 4, 4, 2, 2, 2, 2, 3, 4, 2, 2, 2, 2, 2, 4, 2, 2, 2, 2, 3, 4, 2, 2, 2, 2, 3, 4, 2, 2, 2, 2, 3, 4, 2, 2, 2, 2, 3, 4, 2, 2, 2, 2, 2, 2]
> - Activations: [4, 4, 4, 4, 4, 4, 4, 4, 4, 4, 4, 4, 4, 4, 4, 4, 4, 4, 4, 4, 4, 4, 4, 4, 4, 4, 4, 4, 4, 4, 4, 4, 4, 4, 4, 4, 4, 4, 4, 4, 3, 3, 3, 4, 3, 3, 3, 3]
> - Accuracy: **81.08**
>
> + As shown above, these high-performing configurations reflect the ability of FIT to characterise quantisation.
> + Notably, whilst previous Hessian based methods only use the trace or top eigenvalue to rank and reduce the search space of configurations, our stochastic representation of quantisation indicates that using the Trace*Range gives a far better rank coefficient, especially when including both weights and activations. The inclusion of this noise model is necessary. We reiterate that this is of significant practical importance, as evaluating 5^121 configurations (for every bit of precision and quantization parameter/layer of BERT) is not computationally feasible with any surrogate measure.
> + From a computational perspective, given a batch size of 256, FIT converged with a variance of 0.73 in very few iterations. Conversely, the Hessian trace method was unable to properly converge, even after 200 iterations. This is reflected with a variance of 5256. For this granularity of layer-wise quantization, this renders Hessian-based methods challenging to use in practice.
>
> As we are sure the reviewer will appreciate, performing this many experiments on BERT in such a short timeframe required significant coordination, pushing the limits of our academic resources and network. We would have liked to include the results of Resnet-50 on ImageNet. However, given the short response time, this was not possible so far. We are currently working to access more computing and funding for this purpose, and we hope to include these results in the final revision.

---

> ### Author Response · Authors · 2022-11-11
> **Response to Reviewer QyLp (Part 2)**
>
> **Discussion on the Novelty of our Method:**
>
> Regarding the first point, using the Fisher Information Matrix as a foundation for quantisation has undoubtedly been used before. In the rebuttal revision, we will include the suggested paper in our literature review. Below we clarify where our method differs and where the novelty lies.
> - In this work, we quantise the model using layer-wise mixed precision, and then perform quantisation-aware training (QAT) afterwards. Gao et al. did not use QAT. This retraining process has proved essential in recovering accuracy during the aggressive quantisation schemes that we investigate. [arXiv:2106.08295v1]
> - This combination is entirely novel, and Gao et al. state that they "leave combinations of pruning, model retraining and quantisation like Han et al. (2015a) as future work" (page 8, footer). We show significantly higher accuracies in the figures compared to the basic post-training quantisation setting of https://arxiv.org/pdf/1810.06401.pdf. Our accuracies remain high even for the aggressive quantisation considered [2,3,4,6,8], leading to sub-1 % compression ratios. Hopefully, the empirical setup is now clarified.
> - We maintain a stochastic assumption because of QAT, and that's why FIT is a novel metric. By retaining this stochastic representation, we permit further training after quantisation. In this setup, the exact perturbations $\delta\theta$ are not known a priori, as suggested, and  a quantization noise model is used. Furthermore, maintaining this assumption makes the trace approximation well-founded rather than simply being a quirk of easy computation.
> - Although we understand the intuition behind the suggested baselines for simple quantisation, they break down for the setting we consider, where quantisation and retraining are combined. Furthermore, the recommended baseline scales with the number of layers. For large language models such as BERT, this becomes prohibitively expensive. We are afraid that in debating novelty, the reviewer has yet to consider our computational analysis, which is vital for practitioners.
> - Discussing practical details, whilst we agree that GPUs primarily benefit from 8- and 16- bit precisions, these limitations do not apply to FPGAs optimised for extreme inference times with O(100ns). It is common to scan of bitwidths under various scenarios to effectively trade off accuracy loss and FPGA resource usage [https://arxiv.org/pdf/2101.05108.pdf (Figure 13)].
> - The reviewer mentions non-trivial mixed precision results. As noted in the first revision, we trained hundreds of individual quantisation configurations to compute correlation results. These included mostly non-trivial configurations; a sample of these is highlighted in the rebuttal revision. From the additional empirical analysis in part 1, the importance of these non-trivial configs should be more evident.
>
> **Additional Quantisation Details:**
>
> During our experiments, we employ layer-wise symmetric/asymmetric simulated quantisation. In addition, layer-wise quantisation ranges are accumulated for a short period (30 iterations) at the beginning of QAT to tune the quantisation function across the expected range accurately. Note that this is particularly important for activation quantisation. As a result of the simulated nature of the quantisation, we do not perform batch-norm folding in our experiments. However, this may improve results further, given the additional scaling factor per channel. We will add this discussion to the rebuttal revision appendix.
>
> We hope to alleviate your concerns about our experimental coverage and novelty with this detailed discussion, and we hope you can increase your recommendation score. We are looking forward to further discussion about the novelty of our paper if necessary.
>
> Best regards,
>
> The Authors

---

### Official Review · Reviewer_orGP · 2022-10-27

**Confidence:** 4
**Clarity, Quality, Novelty And Reproducibility:** Please refer to strength and weaknesses.
**Correctness:** 3
**Technical Novelty And Significance:** 2
**Empirical Novelty And Significance:** 2
**Recommendation:** 5

**Strength And Weaknesses:**

Strengths

1. This paper demonstrate clear advantage of FIT to Hessian-based method.

2. The writing of the paper is good.


Weaknesses

1. There are no ImageNet results of the FIT quantization, which is the major weakness of this paper. The author mentioned ImageNet at the beginning of section 4.1, which uses ImageNet to analyze the proposed method, however, no final accuracy results are give,. Only CIFAR-10 and MNIST.

2. The experimental setting and demonstration is unclear. What is the bit width setting for each of the accuracy result?

3. Many baselines regarding mixed precision quantization are not compared. For example: “Mix and match: A novel fpga-centric deep neural network quantization framework” (HPCA 2021), “RMSMP: A Novel Deep Neural Network Quantization Framework with Row-wise Mixed Schemes and Multiple Precisions” (CVPR 2021), and more.




**Summary Of The Paper:**

This paper proposes to use Fisher Information Trace (FIT) to perform mixed quantization of deep neural networks. The author of the paper use FIT to measure sensitivity of parameters and activations regarding quantization, and shows improvement of the performance.

**Summary Of The Review:**

To sum up, the clarity and quality of this paper need to be improved. The author of the paper did some interesting works on model quantization but fails to demonstrate them with thorough experiments. Please refer to strengths and weaknesses for more information.

I think this paper needs to be revised, both on the technical contribution and experiments.

---

> ### Author Response · Authors · 2022-11-11
> **Response to Reviewer orGP (Part 1)**
>
> Dear Reviewer.
>
> Thank you for your thorough review of our work and suggestions for improvement. We are happy that you find the paper well-written and that the method has advantages compared to HAWQ.
>
> **Additional Experiments**
>
> We concur with your recommendation to expand evaluation results. In particular, we analysed BERT on the SST-2 dataset. The rebuttal revision will include changes to the experimental setup, results and discussion sections (the main body and the appendix). For brevity, we summarise this in our response:
> - *Experimental setup:* we chose 50 distinct MPQ configurations for BERT on SST-2 across a range of bit precisions [2,3,4,6,8]. BERT-base is quantised via layer-wise mixed precision simulated asymmetric quantisation of weights and activations.
> - *Results:* We obtain a final correlation score of 0.752 between post-fine-tuning accuracy and FIT score for these MPQ configurations. Similar to other experiments, this demonstrates that FIT has an excellent predictive power of the impact of quantisation, even for large and deep models such as BERT.
> - To demonstrate a use case for FIT, we list below three different MPQ configurations compared to baseline methods constrained by operations (BOPS), showing the superior performance of FIT in selecting high-performing MPQ configurations. Notably, FIT can trade off with any arbitrary hardware constraint (e.g. latency, power), making it very flexible.
>
> *First Baseline Model:*
> - Weights: 4 uniform
> - Activations: 8 uniform
> - Accuracy: 85.66
>
> *Comparable FIT Model [3,4,6,8]*
> - Weights:  [3, 4, 3, 4, 4, 4, 3, 3, 3, 4, 4, 6, 3, 3, 4, 3, 4, 6, 3, 3, 3, 4, 4, 6, 3, 3, 4, 3, 4, 6, 3, 3, 4, 3, 4, 6, 3, 3, 4, 3, 4, 6, 3, 3, 3, 3, 4, 6, 3, 3, 3, 4, 4, 4, 3, 3, 3, 3, 4, 4, 3, 3, 3, 3, 4, 4, 3, 3, 3, 3, 3, 4]
> - Activations: [6, 6, 6, 6, 4, 4, 4, 6, 4, 4, 4, 6, 6, 6, 6, 8, 6, 6, 6, 8, 6, 6, 6, 8, 6, 6, 6, 8, 6, 6, 6, 6, 6, 6, 6, 6, 6, 6, 6, 6, 4, 4, 4, 4, 4, 4, 4, 4]
> - Accuracy: **89.11**
>
> *Comparable FIT Model [4,8]*
> - Weights:  [4, 4, 4, 4, 4, 4, 4, 4, 4, 4, 4, 8, 4, 4, 4, 4, 4, 8, 4, 4, 4, 4, 4, 8, 4, 4, 4, 4, 4, 8, 4, 4, 4, 4, 4, 8, 4, 4, 4, 4, 4, 8, 4, 4, 4, 4, 4, 8, 4, 4, 4, 4, 4, 4, 4, 4, 4, 4, 4, 4, 4, 4, 4, 4, 4, 4, 4, 4, 4, 4, 4, 4]
> - Activations: [8, 8, 8, 8, 4, 4, 4, 8, 4, 4, 4, 8, 4, 4, 4, 8, 8, 8, 8, 8, 8, 8, 8, 8, 8, 8, 8, 8, 8, 8, 8, 8, 8, 8, 8, 8, 8, 8, 8, 4, 4, 4, 4, 4, 4, 4, 4, 4]
> - Accuracy: **90.94**
>
> *Second Baseline Model:*
> - Weights: 2 uniform
> - Activations: 8 uniform
> - Accuracy: 80.62
>
> *Comparable FIT Model [2,3,4]*
> - Weights: [2, 2, 2, 3, 2, 4, 2, 2, 2, 2, 3, 4, 2, 2, 2, 2, 3, 4, 2, 2, 2, 2, 4, 4, 2, 2, 2, 2, 4, 4, 2, 2, 2, 2, 3, 4, 2, 2, 2, 2, 2, 4, 2, 2, 2, 2, 3, 4, 2, 2, 2, 2, 3, 4, 2, 2, 2, 2, 3, 4, 2, 2, 2, 2, 3, 4, 2, 2, 2, 2, 2, 2]
> - Activations: [4, 4, 4, 4, 4, 4, 4, 4, 4, 4, 4, 4, 4, 4, 4, 4, 4, 4, 4, 4, 4, 4, 4, 4, 4, 4, 4, 4, 4, 4, 4, 4, 4, 4, 4, 4, 4, 4, 4, 4, 3, 3, 3, 4, 3, 3, 3, 3]
> - Accuracy: **81.08**
>
> + As shown above, these high-performing configurations reflect the ability of FIT to characterise quantisation.
> + Notably, whilst previous Hessian based methods only use the trace or top eigenvalue to rank and reduce the search space of configurations, our stochastic representation of quantisation indicates that using the Trace*Range gives a far better rank coefficient, especially when including both weights and activations. The inclusion of this noise model is necessary. We reiterate that this is of significant practical importance, as evaluating 5^121 configurations (for every bit of precision and quantization parameter/layer of BERT) is not computationally feasible with any surrogate measure.
> + From a computational perspective, given a batch size of 256, FIT converged with a variance of 0.73 in very few iterations. Conversely, the Hessian trace method was unable to properly converge, even after 200 iterations. This is reflected with a variance of 5256. For this granularity of layer-wise quantization, this renders Hessian-based methods challenging to use in practice.
>
> As we are sure the reviewer will appreciate, performing this many experiments on BERT in such a short timeframe required significant coordination, pushing the limits of our academic resources and network. We would have liked to include the results of Resnet-50 on ImageNet. However, given the short response time, this was not possible so far. We are currently working to access more computing and funding for this purpose, and we hope to include these results in the final revision.

---

> ### Author Response · Authors · 2022-11-11
> **Response to Reviewer orGP (Part 2)**
>
> **Clarity of experimental procedure:**
>
> As noted in the Abstract and Section 4.2, we trained 100’s of quantization configurations to compute correlation results. Consequently, it is not practical to list all of them: a total of 500 individual quantization configurations and accuracy values. We feel it would not improve the clarity of the paper. Instead, we highlight configuration examples for various accuracies and FIT values in Figures 3 and 4. We will include this change in the rebuttal revision.
>
> **Suggested baselines:**
>
> There are many approaches for performing MPQ, and we covered a sample in the literature review. In the paper, we restrict our baseline to layerwise MPQ, commonly done in practice (https://www.nature.com/articles/s42256-021-00356-5). Investigating FIT combined with other more complicated quantization schemes, such as those proposed by the reviewer, provide appealing directions for future work. We will include these suggestions and citations in the discussion section in the rebuttal revision.
>
> With the above changes, we hope to alleviate your concerns about the experimental coverage and clarity and hope you can increase your recommendation score.
>
> Best regards,
> The Authors

---

### Official Review · Reviewer_qJTT · 2022-10-29

**Confidence:** 4
**Correctness:** 3
**Technical Novelty And Significance:** 3
**Empirical Novelty And Significance:** 3
**Recommendation:** 8

**Clarity, Quality, Novelty And Reproducibility:**

- Some of the important analysis seems to be missed: 1) derivation of FIT for activation quantization. 2) detail comparison with the Fisher information matrix  (Li et al., 2021)

**Strength And Weaknesses:**

(Strength)
- A nice overview and derivation of the impact of quantization on the loss.

- The nice properties of the proposed metric are that it is more robust (since it shows a smaller variance) and more efficient (in terms of computation)

- The comparison of correlations that show superior predictability of the proposed methods for the impact of quantization loss.

(Weaknesses)
- The evaluation in Table 2 seems to be very limited. By exploiting the computational efficiency of the proposed algorithm, the authors should also try challenging tasks, such as a large CNN for ImageNet classification or a large language model like BERT.




**Summary Of The Paper:**

This paper proposed a new analytical metric to quantify the impact of quantization for deep neural networks. The proposed metric provides an estimation of quantization impact as accurately as hessian methods yet it is more computationally efficient and its variance is more favorable. The authors compared the proposed method with the other metrics and demonstrate its superior performance.

**Summary Of The Review:**

This paper proposed a practical metric, FIT, for quantitatively analyzing the impact of quantization errors in deep neural networks. FIT's robustness (due to small variance) and computational efficiency would be very useful for the practitioners in this study. However, currently, the evaluation results are very weak; it would be highly appreciated if the authors could include more experimental results, such as ResNet50 on ImageNet.


====== Post rebuttal comments ======
The reviewer thanks the authors for their careful rebuttal. Demonstrating a solid accuracy boost on BERT models seems to make the paper much stronger, and I believe that such a demonstration confirms the applicability of the proposed method for researchers in this field to broader applications. Thus, I increase my score to Accept.

---

> ### Author Response · Authors · 2022-11-11
> **Response to Reviewer qJTT**
>
> Dear Reviewer.
>
> Thank you for your thorough review of our work and suggestions for improvement. We are delighted that you find the proposed metric more robust and efficient than the baseline.
>
> **Additional Experiments**
>
> We concur with your recommendation to expand evaluation results. In particular, we analysed BERT on the SST-2 dataset. The rebuttal revision will include changes to the experimental setup, results and discussion sections (the main body and the appendix). For brevity, we summarise this in our response:
> - *Experimental setup:* we chose 50 distinct MPQ configurations for BERT on SST-2 across a range of bit precisions [2,3,4,6,8]. BERT-base is quantised via layer-wise mixed precision simulated asymmetric quantisation of weights and activations.
> - *Results:* We obtain a final correlation score of 0.752 between post-fine-tuning accuracy and FIT score for these MPQ configurations. Similar to other experiments, this demonstrates that FIT has an excellent predictive power of the impact of quantisation, even for large and deep models such as BERT.
> - To demonstrate a use case for FIT, we list below three different MPQ configurations compared to baseline methods constrained by operations (BOPS), showing the superior performance of FIT in selecting high-performing MPQ configurations. Notably, FIT can trade off with any arbitrary hardware constraint (e.g. latency, power), making it very flexible.
>
> *First Baseline Model:*
> - Weights: 4 uniform
> - Activations: 8 uniform
> - Accuracy: 85.66
>
> *Comparable FIT Model [3,4,6,8]*
> - Weights:  [3, 4, 3, 4, 4, 4, 3, 3, 3, 4, 4, 6, 3, 3, 4, 3, 4, 6, 3, 3, 3, 4, 4, 6, 3, 3, 4, 3, 4, 6, 3, 3, 4, 3, 4, 6, 3, 3, 4, 3, 4, 6, 3, 3, 3, 3, 4, 6, 3, 3, 3, 4, 4, 4, 3, 3, 3, 3, 4, 4, 3, 3, 3, 3, 4, 4, 3, 3, 3, 3, 3, 4]
> - Activations: [6, 6, 6, 6, 4, 4, 4, 6, 4, 4, 4, 6, 6, 6, 6, 8, 6, 6, 6, 8, 6, 6, 6, 8, 6, 6, 6, 8, 6, 6, 6, 6, 6, 6, 6, 6, 6, 6, 6, 6, 4, 4, 4, 4, 4, 4, 4, 4]
> - Accuracy: **89.11**
>
> *Comparable FIT Model [4,8]*
> - Weights:  [4, 4, 4, 4, 4, 4, 4, 4, 4, 4, 4, 8, 4, 4, 4, 4, 4, 8, 4, 4, 4, 4, 4, 8, 4, 4, 4, 4, 4, 8, 4, 4, 4, 4, 4, 8, 4, 4, 4, 4, 4, 8, 4, 4, 4, 4, 4, 8, 4, 4, 4, 4, 4, 4, 4, 4, 4, 4, 4, 4, 4, 4, 4, 4, 4, 4, 4, 4, 4, 4, 4, 4]
> - Activations: [8, 8, 8, 8, 4, 4, 4, 8, 4, 4, 4, 8, 4, 4, 4, 8, 8, 8, 8, 8, 8, 8, 8, 8, 8, 8, 8, 8, 8, 8, 8, 8, 8, 8, 8, 8, 8, 8, 8, 4, 4, 4, 4, 4, 4, 4, 4, 4]
> - Accuracy: **90.94**
>
> *Second Baseline Model:*
> - Weights: 2 uniform
> - Activations: 8 uniform
> - Accuracy: 80.62
>
> *Comparable FIT Model [2,3,4]*
> - Weights: [2, 2, 2, 3, 2, 4, 2, 2, 2, 2, 3, 4, 2, 2, 2, 2, 3, 4, 2, 2, 2, 2, 4, 4, 2, 2, 2, 2, 4, 4, 2, 2, 2, 2, 3, 4, 2, 2, 2, 2, 2, 4, 2, 2, 2, 2, 3, 4, 2, 2, 2, 2, 3, 4, 2, 2, 2, 2, 3, 4, 2, 2, 2, 2, 3, 4, 2, 2, 2, 2, 2, 2]
> - Activations: [4, 4, 4, 4, 4, 4, 4, 4, 4, 4, 4, 4, 4, 4, 4, 4, 4, 4, 4, 4, 4, 4, 4, 4, 4, 4, 4, 4, 4, 4, 4, 4, 4, 4, 4, 4, 4, 4, 4, 4, 3, 3, 3, 4, 3, 3, 3, 3]
> - Accuracy: **81.08**
>
> + As shown above, these high-performing configurations reflect the ability of FIT to characterise quantisation.
> + Notably, whilst previous Hessian based methods only use the trace or top eigenvalue to rank and reduce the search space of configurations, our stochastic representation of quantisation indicates that using the Trace*Range gives a far better rank coefficient, especially when including both weights and activations. The inclusion of this noise model is necessary. We reiterate that this is of significant practical importance, as evaluating 5^121 configurations (for every bit of precision and quantization parameter/layer of BERT) is not computationally feasible with any surrogate measure.
> + From a computational perspective, given a batch size of 256, FIT converged with a variance of 0.73 in very few iterations. Conversely, the Hessian trace method was unable to properly converge, even after 200 iterations. This is reflected with a variance of 5256. For this granularity of layer-wise quantization, this renders Hessian-based methods challenging to use in practice.
>
> As we are sure the reviewer will appreciate, performing this many experiments on BERT in such a short timeframe required significant coordination, pushing the limits of our academic resources and network. We would have liked to include the results of Resnet-50 on ImageNet. However, given the short response time, this was not possible so far. We are currently working to access more computing and funding for this purpose, and we hope to include these results in the final revision.
>
> **Additional changes:** For the rebuttal revision, we rewrote Section 3.2.1 to clarify the derivation of FIT for activation quantisation. We ask that you provide the URL for your suggestion: Fisher Information Matrix (Li et al., 2021). We would like to include it in the rebuttal revision as well.
>
> With the above changes, we hope to alleviate your concerns about the experimental coverage and hope you can increase your recommendation score.
>
> Best regards,
>
> The Authors

---

### Author Response · Authors · 2022-11-16
**Follow-up**

Dear Reviewers,

We would like to thank you all again for your valuable feedback regarding our submission. We have now provided responses and submitted a first rebuttal revision. As the reviewer-author discussion deadline is approaching, we would kindly ask you to let us know if your concerns have been resolved, or if there are outstanding points which require further clarification. We would very much like to include all the reviewers valuable feedback in our revision, including any further discussion!

Thank you very much for your understanding.

The Authors

---

### Public Comment · ~Chen_Tang3 · 2023-02-15
**Citation correctness**

Dear Authors,


Congratulations on the acceptance of FIT to the ICLR 2023! FIT is a very interesting work and I'm happy our work "Mixed-Precision Neural Network Quantization via Learned Layer-Wise Importance" inspires you.

However, FIT did not cite correctly in the current version, and this is a kindly reminder for the correct BibTex code is:
```
@inproceedings{tang2022mixed,
  title={Mixed-Precision Neural Network Quantization via Learned Layer-Wise Importance},
  author={Tang, Chen and Ouyang, Kai and Wang, Zhi and Zhu, Yifei and Ji, Wen and Wang, Yaowei and Zhu, Wenwu},
  booktitle={Computer Vision--ECCV 2022: 17th European Conference, Tel Aviv, Israel, October 23--27, 2022, Proceedings, Part XI},
  pages={259--275},
  year={2022},
  organization={Springer}
}
```



Best,

Chen

---

### Decision · Program_Chairs · 2023-01-20

**Decision:**

Accept: poster

**Justification For Why Not Higher Score:**

As an improved method for the existing setting of mixed-precision network quantization, the scope is probably not broad enough to justify a spotlight.

**Justification For Why Not Lower Score:**

A novel idea with sufficient demonstrated advantages over prior work justifies an accept.

**Metareview: Summary, Strengths And Weaknesses:**

The paper introduces the Fisher Information Trace (FIT) metric as a method of predicting the effects of quantization on a neural network.  This approach is an alternative to prior work on Hessian AWare Quantization (HAWQ) [Dong et al., 2019].  Three reviewers favor accept, one gives a marginally below rating, and one favors reject.  A common request of reviewers (including Reviewer QyLp, who gave an initial reject) was for additional experiments on more challenging settings.  The authors responded with results using FIT to evaluating mixed-precision quantization configurations for BERT models.  Reviewers in favor of accept point to clear development of the idea and demonstrated advantages of FIT in comparison to Hessian-based methods.  Overall, the Area Chair agrees with this plurality of reviewers and also believes the additional results presented in the author response address some of the reviewer concerns on experiments.  The paper might benefit from further expanding experimental settings, such as including CNNs on ImageNet (suggested by Reviewer qJTT).

**Note From Pc:**

if the above contains the word "oral" or "spotlight" please see: "oral" presentation means -> notable-top-5% and "spotlight" means -> notable-top-25%. As stated in our emails, we are disassociating presentation type from AC recommendations